# Description of the embryonic development of *Holothuria floridana* (Pourtalès, 1851) to produce juveniles for aquaculture and restocking

Libni A. Maas-Hernández[1], Miguel A. Olvera-Novoa[1]*, Arlenie Rogers[2]*,
Luis Felaco[3‡], Karina Macal-López[1‡], Teresa Colas-Marrufo[1‡]

**1** Department of Marine Resources, Center for Research and Advanced Studies (Cinvestav), Merida, Yucatán, México, **2** Science Department, University of Belize, Belmopan, Belize, **3** Aquatics Group, Mérida, Yucatán, México

☯ These authors contributed equally to this work.
‡ LF, KM-L and TC-M also contributed equally to this work.
* miguel.olvera@cinvestav.mx (MAON); arogers@ub.edu.bz (AR)

## Abstract

Overexploitation has severely affected the populations of *Holothuria floridana*, the second most valuable species of sea cucumber in the Gulf of Mexico and Greater Caribbean. Knowledge about its reproductive biology is limited, so this study aimed to update particularities about its reproduction and embryonic development that supports aquaculture technologies. Adults were collected in coastal waters off Celestún, Yucatán, Mexico, and transferred to the laboratory to induce spawning by thermal shock. Oocytes were obtained from individual females and fertilized artificially. Samples were incubated under controlled conditions to assess embryonic development using standard histology and by applying optical and scanning electron microscopy techniques. Fertilized oocytes settle and adhere to the substrate through a gelatinous cover. This species exhibits a lecithotrophic embryonic development, with a non-ciliated vitellaria larva that does not go through the auricularia and doliolaria stages, hatching as pentactula. At 1.5 dpf, the primordia of the oral tentacles appeared; for the 2 dpf, the structures of the oral tentacles were apparent, and a primary intestinal tubule was also distinguished, with a link with the anus at 3 dpf when the ring canal of the water vascular system appeared. At 3.5 dpf, the intestinal tubule enlarged and took on a rounded shape, hatching as a pentactula at 4 dpf, with the intestine connected to the mouth and anus, confirming that the digestive tube is complete and functional at this stage showing feeding behavior. The early juvenile stage was attained between 5 and 7 dpf, a much shorter time when compared with planktotrophic species. Its direct embryonic development, the absence of planktotrophic larvae, and the feasibility of inducing spawning through thermal shocks conferred *H. floridana* high potential for aquaculture in the Tropical Western Atlantic. Considering its benthic development, these results give practical information for developing hatchery infrastructure and protocols for rearing in captivity.

**Data availability statement:** All relevant data are within the manuscript and its Supporting information file.

**Funding:** This research was funded by the Ministry of Sustainable Fisheries and Aquaculture of Yucatan, Mexico (Project Grant 2019-SEPASY-ACU-001). LAMH graduate program was supported by Conahcyt scholarship # 776572. The funders had no role in study design, data collection and analysis, where to publish, or manuscript preparation.

**Competing interests:** The authors have declared that no competing interests exist.

## Introduction

The high demand for sea cucumber by the Asian market has resulted in unsustainable fisheries and at-risk, threatened, and depleted wild populations worldwide [1–3]. In the Yucatan Peninsula, Mexico, its fishery began during the first decade of the 21st century, oriented to the capture of mainly *Isostichopus badionotus* (Selenka, 1867) and *Holothuria floridana* (Pourtalès, 1851). Weak compliance with management measures, coupled with illegal, unreported, and unregulated fishing (IUU), resulted in a rapid collapse of the sea cucumber populations that resulted in a permanent ban on the capture of all species as of 2018 [4–8].

Overfishing sea cucumbers has led to declining or disappearing wild populations, making aquaculture an alternative to meet its commercial demand [9]. Although technology has been developed to culture commercially relevant species such as *Apostichopus japonicus* and *Holothuria scabra*, this activity only partially satisfies market demand [10–12]. In this sense, there is a need to diversify the production of sea cucumbers, particularly with species from tropical waters, for which more research is required to update and expand information that supports technological developments for their culture, including aspects of basic biology, in particular, those related to their reproduction and the production of juveniles in captivity while resolving technological, economic and social factors [3,9,10,13].

*Holothuria floridana* is distributed on the Central Western Atlantic, from Florida to Brazil, including the Gulf of Mexico and the Greater Caribbean. It inhabits 0 to <20 m deep coastal areas in muddy and sandy bottoms, grasslands, and reefs [13,14]. They are gonochoric, without sexual dimorphism, and reproduce in the Yucatan Peninsula, Mexico, during spring-summer months [15,16]. During spawning, they broadcast the gametes into the water column, and the fertilized oocytes settle and adhere to available substrates, including the body surface of the progenitors, where embryonic development occurs [17]. They have a fast lecithotrophic embryonic development within the fertilization envelope as an unfed vitellaria larva [17–19], hatching as a pentactula larva.

Due to its strong resilience and ease of reproduction under controlled conditions, this species has a high potential for aquaculture [14]. It is, therefore, imperative to critically evaluate the existing knowledge related to its reproductive patterns and fill gaps with updated information, especially in its early life cycle stages. This need is underscored by the fact that the limited available information on its larval development is from studies conducted at the beginning of the twentieth century [20]. However, a concise description of its reproductive season and gonadic maturation developmental cycle was made for this species in the Campeche Sound [16], the same area where broodstock was collected for this study. Accordingly, female gonads showed asynchronous oocyte maturation [16], indicating a capacity to undergo several spawning during the breeding season [14,16]. Therefore, the objective of this work was to describe the embryonic development of *H. floridana* using current histology and microscopy techniques to generate practical information for the design of hatchery protocols and facilities for its reproduction and rearing, oriented to the production of juveniles for aquaculture and restocking purposes.

## Materials and methods

### Collection of *H. floridana*

*Holothuria floridana* broodstock was caught by scuba diving during March and May 2022 in coastal waters off Celestún, Yucatán, Mexico, with the authorization and collaboration of the personnel from the Mexican Institute for Research in Sustainable Fisheries and Aquaculture. The adults were transported in ice coolers to the Cinvestav marine station in Telchac Puerto, Yucatán, where they were maintained in fiberglass tanks in a closed recirculating system until spawning.

### Histology of female gonads

The degree of gonadic maturation was determined according to [16], using the gonads of two females fixed for two days in Bouin's solution, preserved in 70% ethanol, and dehydrated in increasing concentrations of ethanol, treated with CitriSolv and included in Paraplast [21]. Histological slices of 6 μm were stained with hematoxylin-eosin (HE) [22]. In the slices, the number of oocytes was compared in five randomly selected $2\,mm^2$ fields from the long and short tubules and the anterior, middle, and posterior sections of each type of tubule, for a total of 30 fields (5 fields x 3 zones x 2 types of tubules). Digital images were used for the estimation of the number of oocytes *per* field (Leica DM 2700M microscope; Leica MC170 HD camera, LAS EZ Leica Application Suite Version 3.4.0, Wetzlar, Germany), and the images were processed by spectral analysis and binary segmentation using Image-Pro-Plus 6.0 software (Media Cybernetics, Inc., Rockville, MD). All mature or secondary growth oocytes (cortical alveoli oocytes, vitellogenic oocytes, mature oocytes [see 23]) only were counted; reserve or primary oocytes and oocytes identified as degenerated (i.e., atresia) were eliminated. Oocytes that overlapped at the edges of the fields were counted, since the fields analyzed were not adjacent to each other, to avoid errors of counting the same oocyte twice in two different fields.

### Spawning induction and oocyte handling

Spawning was induced using an in-house thermal shock protocol developed for *H. floridana*. The initial water temperature was 27–28°C and was lowered by 4–6°C. Once the desired temperature was reached, the broodstock was placed first in the cold water for 30–45 min; after this, individuals were transported to a second set of tanks that had previously been heated 4–6°C higher than the initial temperature; there, the organisms spend 30–45 min again. Males initiated spawning behavior by releasing sperm into the water column. When females exhibited spawning behavior, each one was removed and placed in individual containers with water from the same tank, enabling the collection of the oocytes from each female separately. This process allowed for the counting of the exact number of oocytes per female. Spawning inductions were repeated thrice the same night until the desired number of oocytes was attained.

The oocytes for this study were collected with a beaker and placed in a 2 L container with seawater from the same tank. Subsequently, 20 mL of seawater with sperm collected directly from the males' gonopore was added and with gentle aeration. After 2 h, the fertilized oocytes were transferred to 5 L plastic jars with an approximate density of one oocyte $mL^{-1}$. The jars were maintained in a 10 L water bath tray at $26 \pm 1°C$, and samples were collected at regular intervals to document the embryonic development. Every 24 hours, 100% of the seawater in the 5 L jars was changed. The resulting pentactula and juveniles were fed live microalgae (*Chaetoceros* sp. and *Nannochloropsis* sp.).

### Embryo cultures, fixation, and imaging

The development of embryos, larvae, and juveniles was observed every 30 minutes during the first six hours and then every two hours until 12 hours post fertilization (hpf). Then, every 12 hours from day 1 to day 6, every 24 hours from day 7–10, and at 15, 20, 30, 45, and 60 days post fertilization (dpf). Each time, samples of embryos, larvae, and juveniles were placed in separate vials containing 2.5 mL of seawater and 2–3 drops of clove oil to relax their bodies and fix them in 10% formaldehyde for histology. The samples were observed and photographed using the LEICA analysis system (LAS

ES ver 3.0, Leica Microsystems, Wetzlar, Germany). The length of larvae and juveniles was obtained using Motic Image Plus 3.0 software (Motic, Xiamen, China).

The external morphology of embryos and juveniles was documented by scanning microscopy, for which samples preserved in formaldehyde were washed twice with phosphate-buffered saline (PBS), kept for one hour in 2.5% glutaraldehyde and sodium cacodylate ($C_2H_6AsNaO_2$) 0.1M (pH 7.2) and washed again with PBS, followed by gradual dehydration in ethanol (10%, 50%, 70%, 90% and 100%); after that, they were wrapped in filter paper to be desiccated with $CO_2$ (Quorum 850 Critical Point Dryer, Quorum Technologies, U.K.), metalized with gold and palladium for 50 s (Quorum Q150R ES) and scanned on a Jeol-7600F (JEOL Ltd. Japan) electron microscope operated at 5.0 kV.

## Histological analysis

To describe organogenesis, oocyte samples, larvae, and juveniles previously fixed in formaldehyde were preserved for seven days in Bouin's solution [21]. Subsequently, they were washed with 70% ethanol, encapsulated in 4% agarose, wrapped in filter paper, and placed in histological cassettes for dehydration with increasing concentrations of ethanol and chloroform in an automatic processor (Kedee; Jinhua, Z.J.), then embedded in paraffin following the procedure established by [24], and were transverse-sectioned at 5 μm with a Minot-type rotation microtome (Kedee; Jinhua, Z.J.). The histological slides were stained with hematoxylin-eosin (HE) [24]. The lamellae were observed and photographed using the LEICA system.

## Statistical analysis

After analysis of the data's normality and homoscedasticity, descriptive statistics were accordingly performed on the measurements of oocytes, larvae, and juveniles (InfoStat, UNC, Argentina). The gonadic development was analyzed using Pearson's Chi-square goodness-of-fit test to identify whether oocyte density frequencies were uniform among sections of tubules (anterior, middle, and posterior), tubules (short and long), and females. The oocyte density frequencies (number of oocytes mm$^2$) observed in each section and each tubule of the ovary were analyzed through contingency tables (rows x columns) at three levels for each female: 1) between the anterior, middle and posterior sections of each tubule; 2) between the short and long tubules of each anterior, middle and posterior section and, 3) between the short and long tubules, Bonferroni's post hoc analysis was applied when heterogeneity in the distributions was determined. The replicated G goodness test [25] was applied to confirm heterogeneity in the different sections of the tubules using the libraries FSA [26] and DescTools [27] of R [28]. A type I error was considered for these statistical analyses (α = 0.05).

## Results

### Histology of female gonads

The two females used to determine gonadic maturity and type of reproduction had an average compressed length of 22.75 ± 1.06 cm and a body weight of 241.23 ± 13.39 g (mean ± S.D.). The ovaries, with a bright orange color (Fig 1A), weighed, on average, 52.94 ± 4.78 g. Characteristically, it was observed that ovaries were formed by a series of skein-like long and short tubules, with the longest being the ones with the greatest branching (Fig 1B).

Oocytes were observed in the short and long tubules and any other gonad sections of both females. Considering the number and distribution of mature oocytes present in the short or long gonad tubules and the gonad sections (Fig 1C), it was determined that female 1 was in Stage III: "Mature" especially since the oocytes were densely packed within the ovarian tubule, leading to irregular shapes due to spatial compression. This high degree of crowding is a typical histological indicator of sexual maturity in females, reflecting advanced vitellogenic or mature stages of oogenesis. Female 2 (Fig 1D) was classified as Stage II: "Development" and presented a relatively empty lumen with a few mature oocytes, degrading oocytes, and several primary oocytes lining the tubule, likely serving as reserves for future clutches.

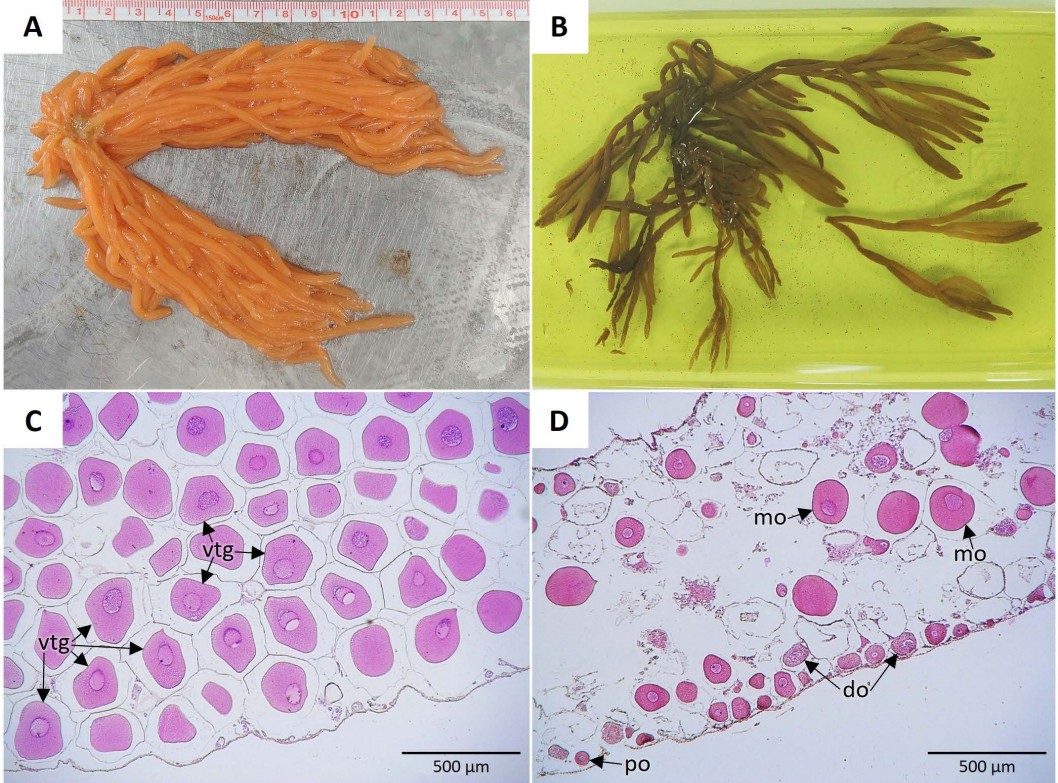

**Fig 1.** *Holothuria floridana* **ovaries (A, B) and histological sections of the tubules (C, D).** The gonad fixed for histology **(B)** shows its structuring in long and short-branched tubules. **(C)** Female 1 mature ovary tubule with the lumen filled with vitellogenic oocytes (vtg). **(D)** Female 2 developing ovary tubule with degrading oocytes (do), a few mature oocytes (mo), and primary oocytes (po) lining the tubule.

Table 1 shows the results of the density and frequency of oocytes in the ovary of the two females histologically analyzed to confirm the degree of maturity of the broodstock used in the study. A significant difference was observed between the density of oocytes in the short and long tubules (Female 1, $x^2 = 4.11$, df 1, $P = 0.0425$; Female 2, $x^2 = 35.94$, df 1, $P < 0.0001$). The average number of oocytes per 2 mm$^2$ was lower in the long tubules, with 176 ± 42 in female 1 and 126 ± 17 in female 2, compared with the short tubules that presented 199 ± 6 oocytes in female 1 and 188 ± 20 in female 2 (Mean ± S.D.).

A heterogeneity in the number of oocytes was observed among the sections of the long tubules in female 1 (number of oocytes = 528, $x^2 = 19.95$, df = 2, $P = 0.0001$; Bonferroni-$P = 0.0004$) compared to that of female 2 (number of oocytes = 379, $x^2 = 4.69$, df = 2, $P = 0.095$; Bonferroni-$P = 0.3832$), Table 1. The number of oocytes in the long tubules of female 1 (n = 130) was significantly lower in the anterior zone (G = 13.09; d.f. = 1, $P < 0.05$) compared to that observed in the middle zone (n = 186, G = 1.02, d.f. = 1, $P = 0.31$) and posterior (n = 212, G = 0.71, d.f. = 1, $P = 0.39$). The pooled result of this test indicated significantly different heterogeneity in oocyte density between short and long tubules and between the different sections (G = 4.12, d.f. = 1, $P < 0.05$).

In the case of female 2, although no variations were observed in the frequency of oocyte density among sections of each type of tubule, the G test identified a significantly different heterogeneity in the number of oocytes between tubules for the anterior and posterior areas, with a lower number of oocytes in the long tubules (anterior: n = 114, G = 23.38, d.f. = 1, $P < 0.05$; Medium: n = 146, G = 1.16, d.f. = 1, $P = 0.28$; Posterior; n = 119; G = 20.34, d.f. = 1, $P < 0.05$), than the number observed at the

**Table 1. Chi-square goodness-of-fit and Bonferroni correction for oocyte density frequencies in short and long ovary tubules and in anterior (A), middle (M), and posterior (P) sections of ovary tubules in two females of *Holothuria floridana*.**

| Female | Ovary | | Oocyte density | | Chi-square[1] | | | |
|---|---|---|---|---|---|---|---|---|
| | Lobe | Section | Frequency[a] | Total[a] | $x2$ | d.f. | *P* | *Bonferroni-P* |
| No. 1 | Short | A | 195 | 596 | 0.41 | 2 | 0.8163 | 1.0000 |
| | | M | 206 | | | | | |
| | | P | 195 | | | | | |
| | Long | A | 130 | 528 | 19.95 | 2 | 0.0001* | 0.0004* |
| | | M | 186 | | | | | |
| | | P | 212 | | | | | |
| No.2 | Short | A | 199 | 563 | 4.11 | 2 | 0.1283 | 0.5132 |
| | | M | 165 | | | | | |
| | | P | 199 | | | | | |
| | Long | A | 114 | 379 | 4.69 | 2 | 0.0958 | 0.3832 |
| | | M | 146 | | | | | |
| | | P | 119 | | | | | |
| No. 1 No. 2 | Short | | 596 563 | 1159 | 0.94 | 1 | 0.3324 | 0.6648 |
| No. 1 No. 2 | Long | | 528 379 | 907 | 24.48 | 1 | <0.0001 | 0.0002* |

[a] = number *per* unit area.

[1] d.f. = degrees of freedom (r-1) (c-1) where r = ovary tubules, and c = oocyte sections analyzed. Total area analyzed per section = 2 mm$^2$.

anterior and posterior sections of the short tubules. The pooled result of the G test indicated significantly different heterogeneity in oocyte density between short and long tubules and among the different sections (G = 36.17, d.f. = 1, *P* < 0.05).

When comparing the frequencies of oocyte density between females for the short and long tubules, the Chi-square test evidenced a heterogeneity in the number of oocytes between the long tubules of the two females. This result was confirmed by Bonferroni correction (long tubule: female 1, n = 528 oocytes; female 2, n = 379 oocytes; $x^2$ = 24.48, d.f. = 1, *P* < 0.0001, Bonferroni-*P* = 0.0002. Short tubule: female 1,596 oocytes; female 2, 563 oocytes; $x^2$ = 0.94, d.f. = 1, *P* = 0.33, Bonferroni-*P* = 0.66), Table 1.

## Spawning induction and oocyte density

During the spawning induction, the water temperature in the broodstock tanks was 27.5–28°C, so the minimal and highest temperatures applied for thermal shock were 20–22°C and 32–33°C, respectively. Serial thermal shock (cold-hot water) induced spermiation starting from the second repetition, with spawning occurring 30–40 min after the third repetition, in both cases with the higher temperature. The oocytes described in embryonic development correspond to a female spawning that released 11,875 oocytes.

## Embryonic development

The fertilization envelope that covered the oocytes (Fig 2A) was surrounded by an adhesive gelatinous capsule (Fig 2A–D). The oocyte protuberance became visible (Fig 2B) as well. After fertilization, the diameter of the oocytes was 394 ± 16 µm (n = 20; mean ± S.D.), while the thickness of the adhesive gelatinous capsule was 160.54 ± 37.63 µm (n = 20; mean ± S.D.), which gradually decayed as embryonic development progressed until it reached a minimum thickness of 83.59 ± 14.08 µm (n = 27; mean ± S.D.) at hatching.

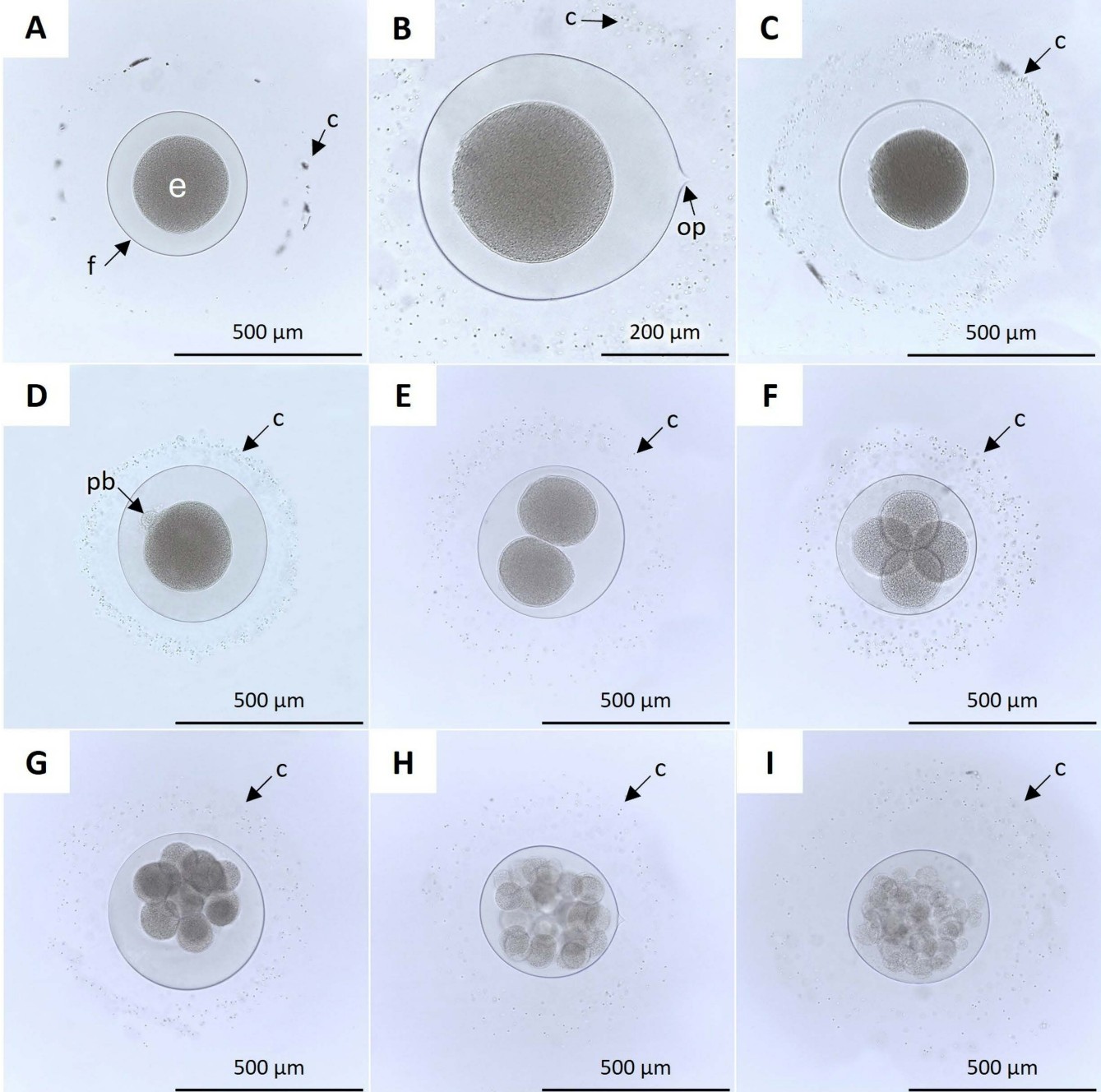

**Fig 2. Embryonic development of *H. floridana* from fertilization to morula [ 23]: A)** fertilization envelope (f) and fertilized one-cell stage embryo **(e). B)** adherent gelatinous capsule (c) and oocyte protuberance (op). **C)** adherent gelatinous capsule (c). **D)** Extrusion of the polar body (pb). **E)** First cell division until the formation of the two blastomeres (2.5 hpf); **F)** Second cell division with four blastomeres (3 hpf); **G)** Third cell division (4 hpf); **H-I**: Morula at 5 hpf.

A polar body was extruded one-hour post fertilization (hpf) (Fig 2D). The first cell division occurred between 2 and 2.5 hpf (Fig 2E). The second cell division occurred between 2.5 and 3 hpf (Fig 2F). The subsequent cell divisions occurred approximately every 30 min, observing 16 blastomeres by 4 hpf. (Fig 2G). Finally, the morula was observed at 4–5 hpf (Figs 2H–I).

The blastula, characterized by a fluid-filled cavity called blastocoel, forms around 8 hpf (Fig 3A, B), and invagination, the inward movement of tissues into the blastocoel, signifies the onset of gastrulation, which occurred between 12–24 hpf, with the gastrula fully formed at 24 hpf (Fig 3C). At 1.5 days post fertilization (dpf), the lecithotrophic vitellaria larva was observed inside the fertilization envelope, showing rounded edges on the opposite side of the blastopore as the primordia of the primary oral tentacles (Fig 3D). At 2 dpf, the rounded edges showed early differentiation into the primary oral tentacles (Fig 3E), which continued to develop and reached greater structural definition at 3 dpf (Fig 3F, G). At 3.5 dpf, the primary oral tentacle gradually extended, intermittently protruding and retracting, eventually rupturing the fertilization envelope, facilitating the pentactula larvae to hatch (Fig 3H, I).

From day 4 post-fertilization, the pentactula presented a main podium in the posterior region of the body, which elongated as development progressed (Fig 4A–C). The intestine was visible at 6 dpf (Fig 4C), and by 7 dpf, a secondary podium was observed at the center of the body, marking the transition to the early juvenile stage (Fig 4D). At 10 dpf, the juvenile had two secondary podia and a visible intestine. However, until 20 dpf, the overall morphology remained unchanged, aside from an increase in size and the progressive development of the intestine (Fig 4E), including further development of the sigmoid-shaped intestine and the anus (Fig 4F). At 30 dpf, in addition to the main podium, the juvenile had nine secondary podia in three rows with three podia each (Fig 4G). By the 45 dpf, juveniles had six oral tentacles, the main podium, and ten secondary podia (Fig 4H). At 60 dpf, the juvenile had eight oral tentacles (Fig 4I). Table 2 summarizes the biometric data of the pentactula and juveniles at different developmental times.

Developing biscuit ossicles were observed at 5 dpf (Fig 5A), forming towers with four circles at the base around the 7 dpf (Fig 5B). By 9 dpf, the ossicle towers reached greater height, and an additional row of circles surrounded the four central circles (Fig 5C).

The scanning electron microscopy (SEM) image of *H. floridana* oocyte shows a spherical shape at 1.5 hpf (Fig 6A), and details of the fertilization envelope are observed. The rupture of the fertilization envelope allowed the observation of the vitellaria, which has a rough appearance without apparent cilia (Fig 6B). At 5.5 dpf, a pentactula was observed with a smooth body and short oral tentacles without development (Fig 6C). The SEM image of the juvenile at 45 dpf (Fig 6D) shows the ossicles protruding out of the body's surface (Fig 6E) and the bud of the oral tentacle with a rough surface (Fig 6F, G). Fig 6H corresponds to a juvenile at 60 dpf, where the ossicles (Fig 6I), podia (Fig 6J), and eight oral tentacles with peltate buds (Fig 6K, L) are observed.

## Embryonic development and organogenesis

The histology of samples post-fertilization allowed the observation of the organogenesis of the developing embryo and the vitellaria at the different sampling times, from the blastomeres that make up the morula (Fig 7A) to the most developed structures. In the sections obtained at 12 hpf, blastulae (Fig 7B) and blastulae in the process of gastrulation were observed (Fig 7C), and the conformation of the gastrula at 24 hpf (Fig 7D). At 1.5 dpf, the primordia of the oral tentacles were observed (Fig 7E), and at 2 dpf, the structures of the primary oral tentacles showed greater definition; an isolated primary intestinal tubule (primitive gut tube) was also distinguished (Fig 7F).

Between 2 and 4 dpf, the digestive system became more apparent and showed a high level of organization. At 2.5 dpf, the closed intestinal tubule did not show an evident connection (Fig 8A). At 3 dpf, the intestinal tubule showed a connection with the anus, and the ring canal of the water vascular system appeared (Fig 8B). At 3.5 dpf, the anterior part of the intestinal tubule enlarged and assumed a rounded shape, leading to the formation of the stomach. An increase in the development of the ring canal of the water vascular system was also observed (Fig 8C), coinciding with the onset of the

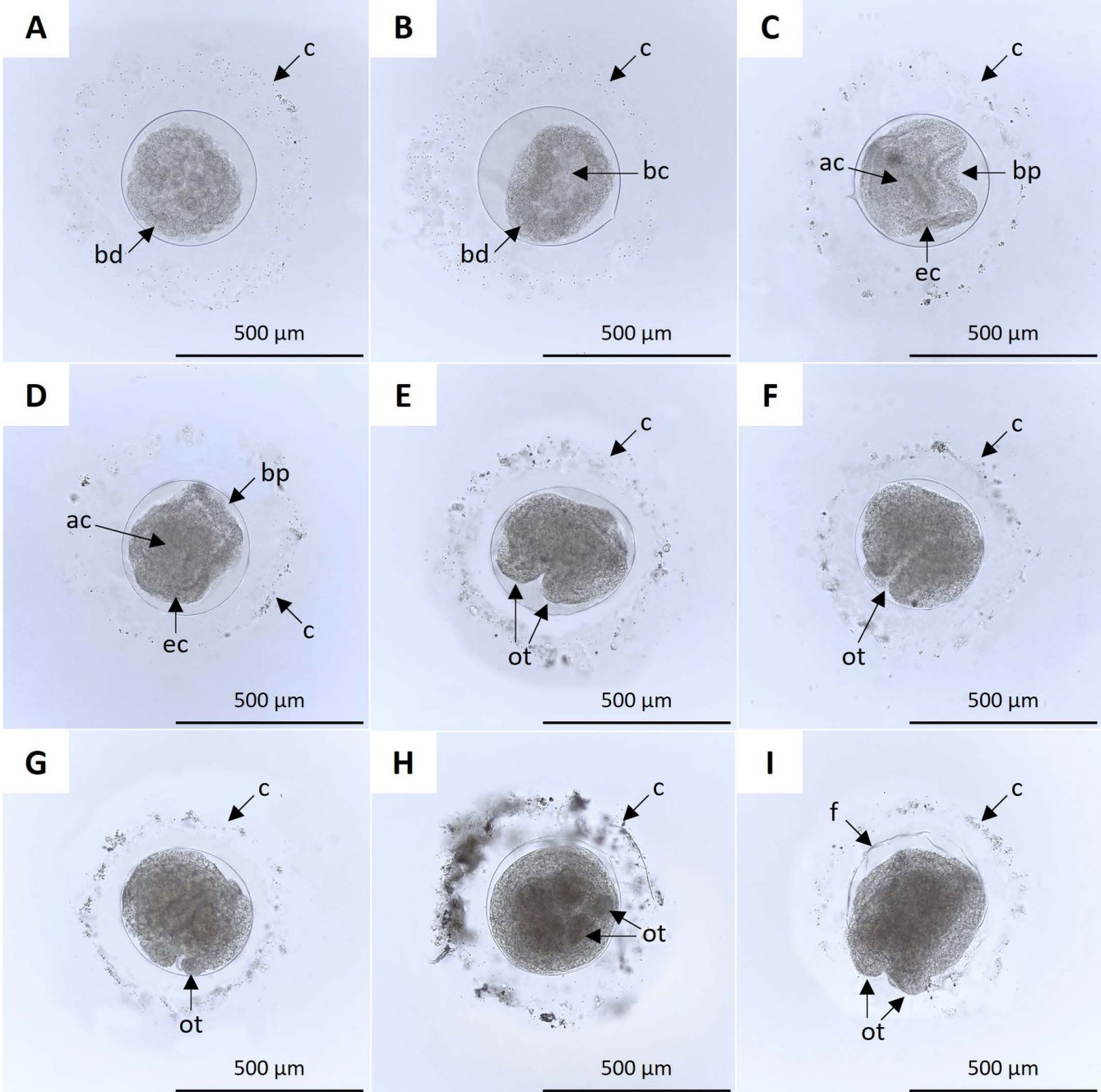

**Fig 3. Embryonic development of *H. floridana* from 8 hpf to 3.5 dpf. A)** Blastula: blastoderm (bd). **B)** Blastula, characterized by the blastocele (bc). **C)** 24-h gastrula: blastopore (bp), archenteron (ac), and ectoderm (ec). **D)** 1.5 dpf vitellaria larva showing rounded edges on the opposite side of the blastopore, indicating the primordia of the primary oral tentacles. **E)** 2-dpf larva with early differentiation of primary oral tentacles (ot). **F)** 2.5 dpf vitellaria larva with further differentiation of oral tentacles. **G)** Larva vitellaria of 3 dpf with defined oral tentacles. **H)** Front view of 3.5 dpf vitellaria larva with five oral tentacles. **I)** Initiation of hatching at 3.5 dpf with the rupture of the fertilization envelope (f) by the oral tentacles.

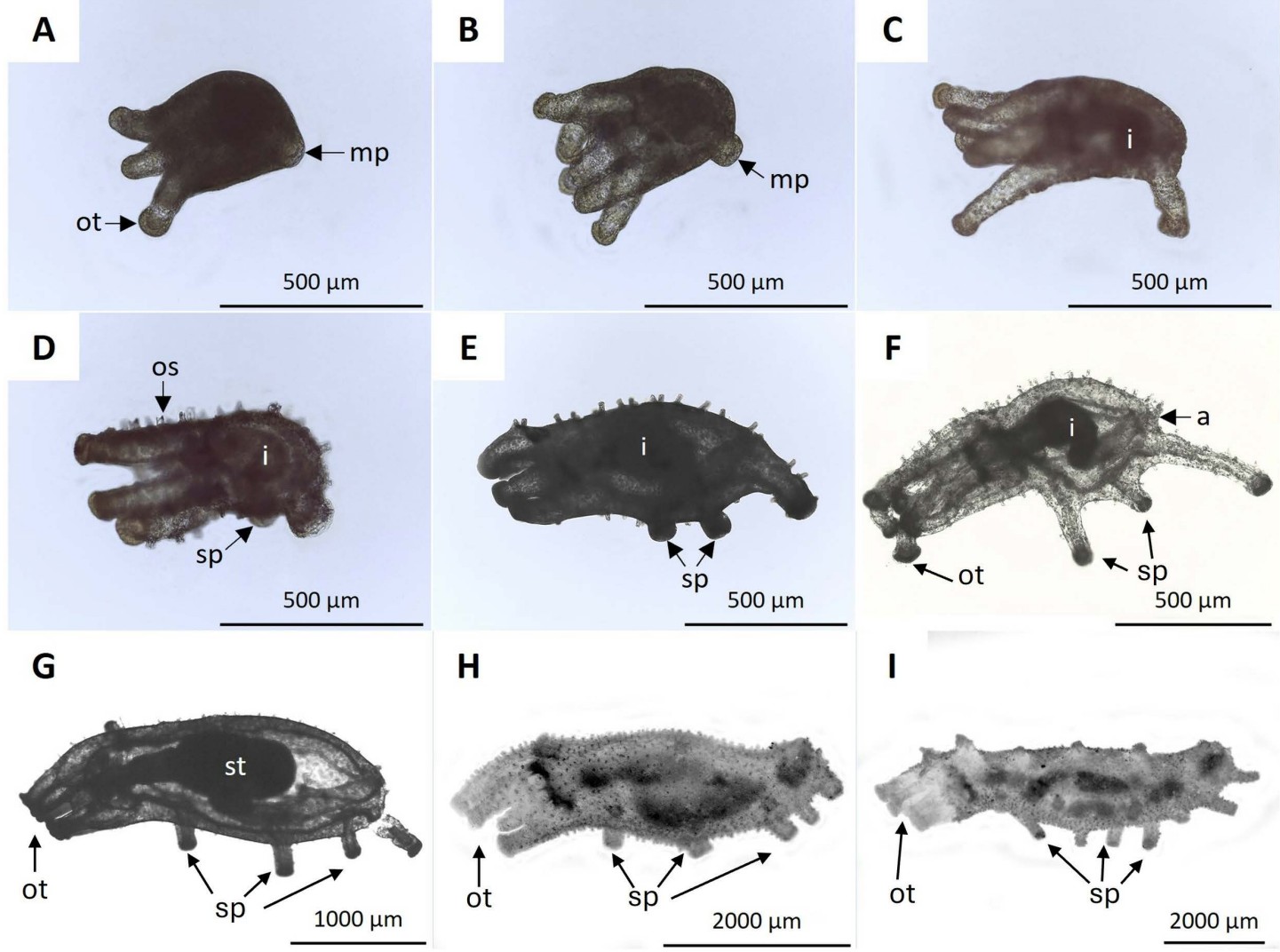

**Fig 4. Pentactula and juvenile development of *H. floridana*. A)** Pentactula at 4 dpf with oral tentacles (ot) and main podium (mp). **B)** 5 dpf pentactula larva: five oral tentacles and main podium. **C)** Pentactula with 6 dpf: sigmoidal intestine (i). **D)** Juvenile at 7 dpf with a secondary podium (sp), ossicles (os), and intestine. **E)** Juvenile of 10 dpf with two secondary podia and intestine. **F)** Juvenile of 20 dpf: sigmoid intestine and anus (a). **G)** Juvenile with 30 dpf showing main podium and three rows of podia, each containing three podia: stomach (st). **H)** Juvenile of 45 dpf showing six oral tentacles, main podium, and 10 secondary podia. **I)** Juvenile of 60 dpf showing eight oral tentacles.

hatching of the pentactula. At 4.5 dpf, the digestive tract acquired a sigmoidal shape, with the intestine connected to the mouth and anus (Figs 8D–F). At 5.5 dpf, a high cell concentration was observed in the oral region of the pentactula, and the ring canal was longer (Fig 8G). At 6 dpf, the digestive system appeared as a continuous structure, distinguishing the esophagus, stomach, and intestine ending with the anus (Figs 8H, I).

Regarding the development of the digestive system of early juveniles, at 9 and 10 dpf, the esophagus, stomach, intestine, and anus were evident (Fig 9A, B). At 15 dpf, the esophagus and intestine showed circular folds (Fig 9C). At 20 dpf, juveniles exhibited a distinct elongated and granular structure, identified as the respiratory tree, which extended from the anus along the edge of the coelom (Fig 9D). In juveniles at 30 and 45 dpf, the esophagus and stomach were contiguous,

**Table 2. Biometric data of pentactula and juveniles of *H. floridana* on different days post fertilization (dpf).**

| Developmental stage | dpf | n | Length (µm) | |
|---|---|---|---|---|
| | | | **Mean** | **±S.D.** |
| Pentactula | 4 | 16 | 431.36 | 62.53 |
| Early juvenile | 5 | 5 | 496.34 | 35.99 |
| Juvenile | 10 | 22 | 655.29 | 89.28 |
| " | 20 | 12 | 1,054.31 | 63.72 |
| " | 30 | 14 | 2,661.75 | 349.11 |
| " | 45 | 10 | 3,559.87 | 472.88 |
| Juvenile | 60 | 10 | 6,233.36 | 1715.38 |

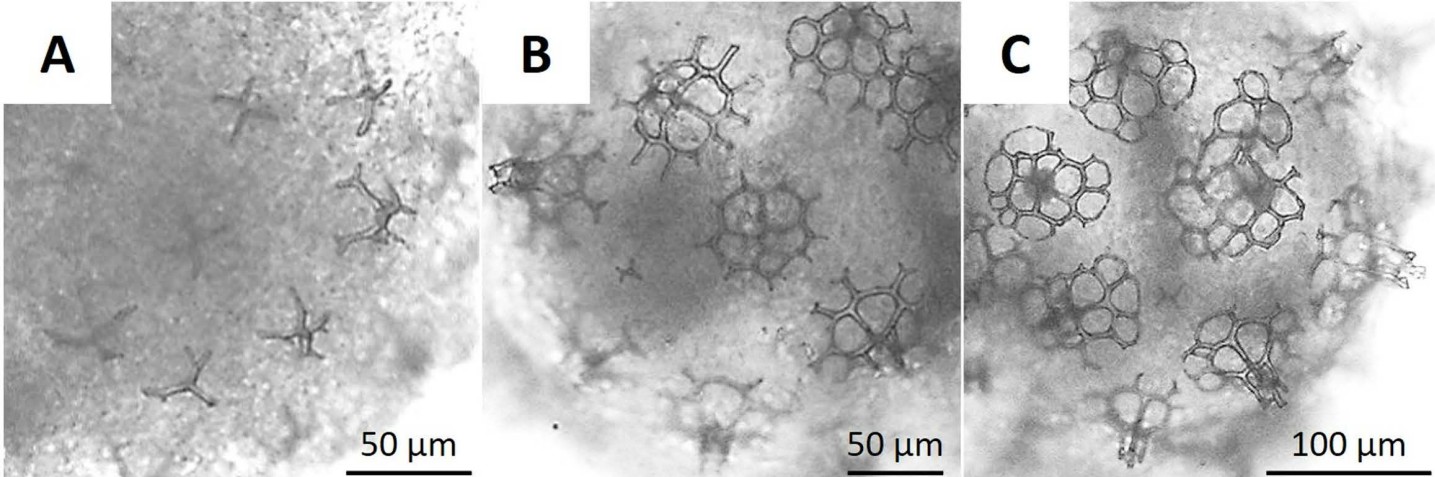

**Fig 5. Biscuit ossicles of *H. floridana* at five (A), seven (B), and nine days (C) post-fertilization.**

and the intestine was elongated. As in the case of adults, the section of the intestine corresponding to the cloaca was identifiable. Details of the podia and the pelted oral tentacles were also observed (Figs 9E, F).

## Discussion

The breeding season in tropical holothuroids can vary widely, with more continuous or longer spawning seasons throughout the year [29]. *Holothuria floridana* spawns year-round in the Bay of Campeche, on the western coast of the Yucatan Peninsula, Mexico, with a significant peak from March to May and a secondary peak from August to September [16]. The broodstock used in this study came from the same geographic region and was induced for spawning during the reproductive period of the species.

In the Bay of Campeche, the size at first sexual maturity for this species is 13.4 cm [16]; in Belize, spawning was obtained with individuals between 11 and 35 cm [17]. In this study, the average compressed length of the broodstock was 22.75 cm, which is considered appropriate for breeding.

Histological observations of the gonads of two females, sampled to confirm maturity and reproductive suitability of the collected broodstock, revealed oocyte development that allowed classification of the females into two of the five reproductive phases: I, undifferentiated or resting; II, development; III, maturity; IV, spawning, and V, post-spawning or spawned. These phases are commonly used to describe gonadal development in holothuroids, such as *H. fuscogilva* [30], *H. scabra*

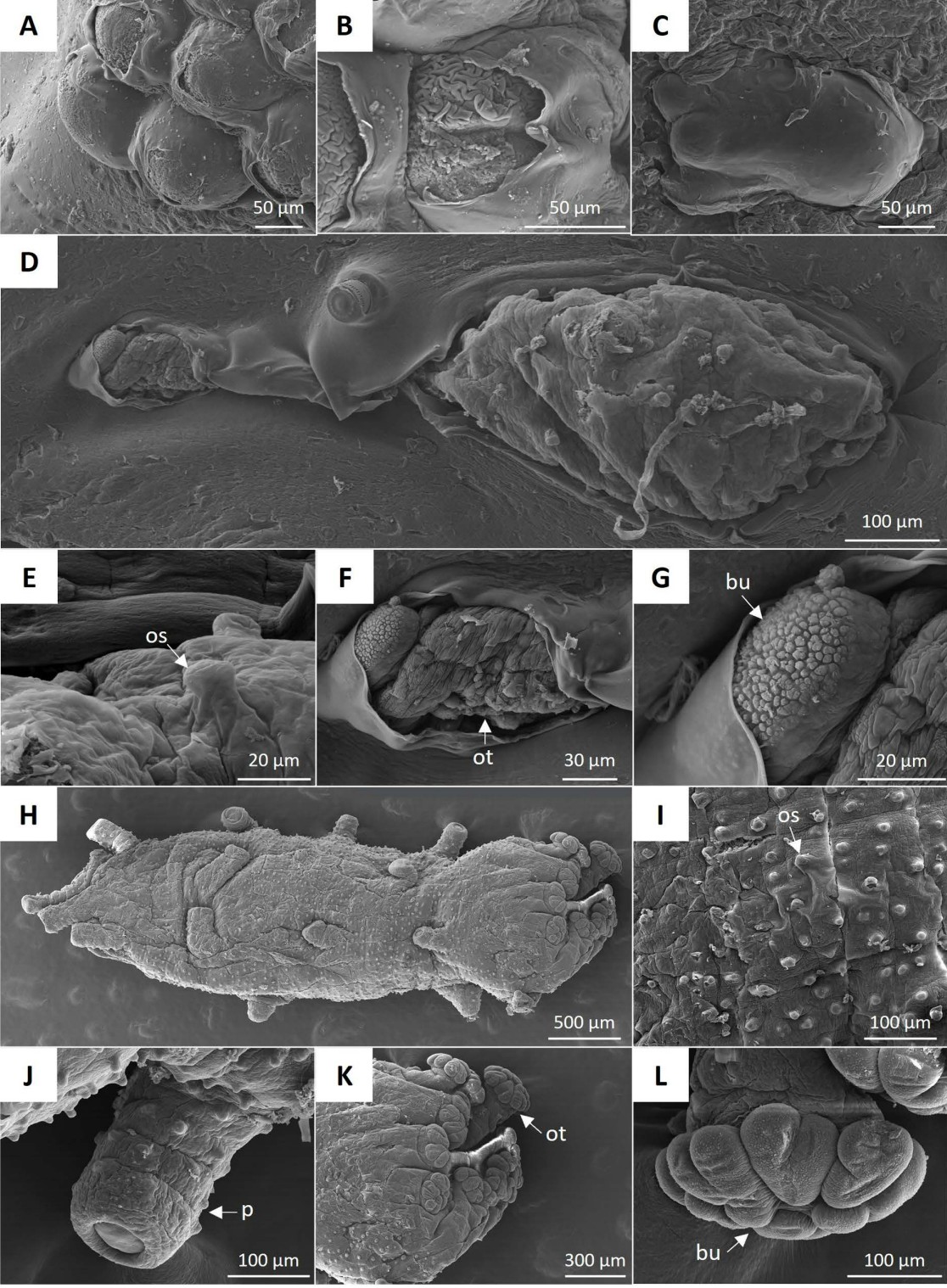

**Fig 6. SEM images of *H. floridana*. A)** Oocytes showing spherical shape and details of the broken fertilization envelope. **B)** Details of the vitellaria with a rough surface. **C)** 5.5 dpf pentactula. **D)** view of the juvenile's body at 45 dpf partially embedded in agarose. At this time, it is possible to distinguish **E)** ossicles (os), **F)** oral tentacle (ot), and **G)** oral tentacle bud (bu). At 60 dpf, the entire juvenile's body is observed **(H)** as well as the **I)** ossicles (os), **J)** podia (p), **K)** oral tentacles (ot), and **L)** peltate buds of the oral tentacle (bu).

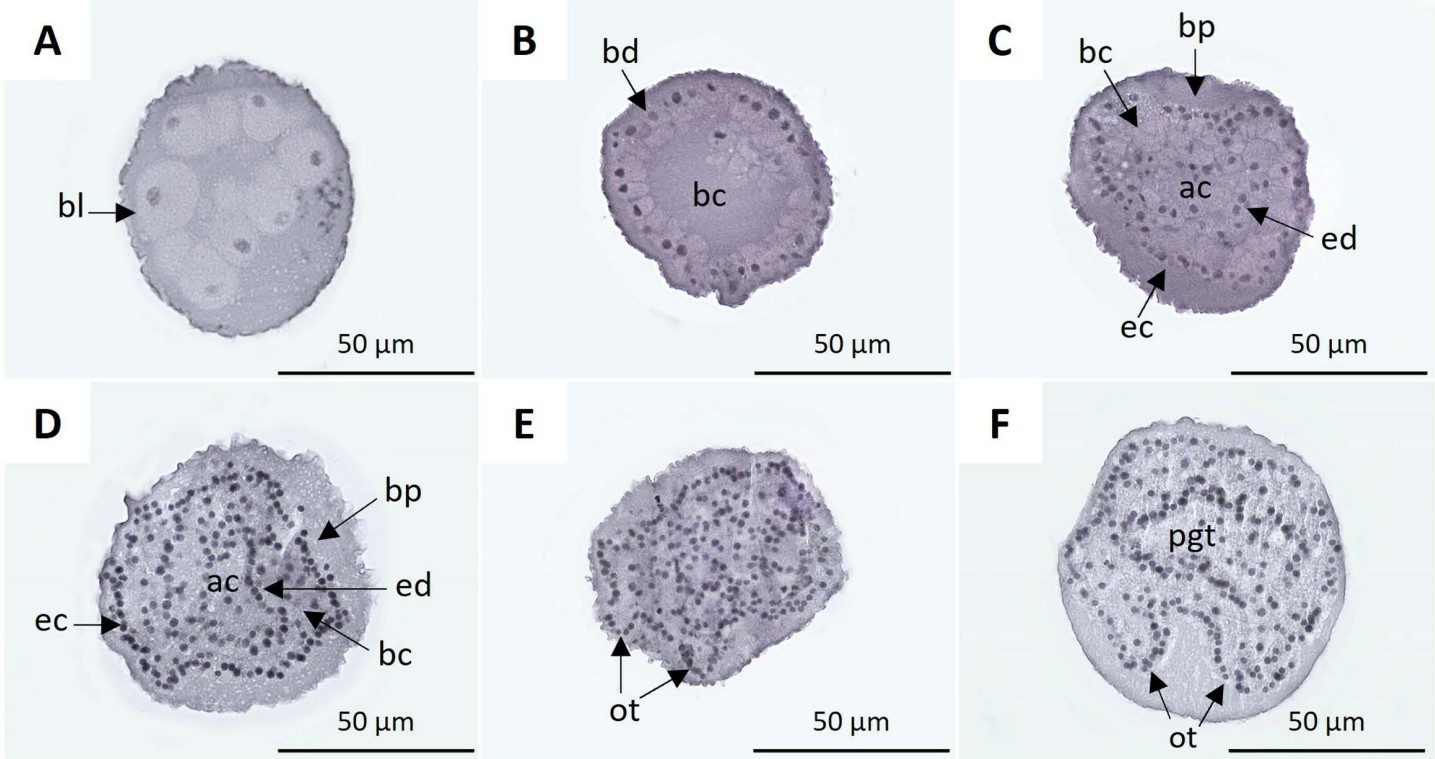

**Fig 7. Histological sections of *H. floridana* embryos. A)** The morula in the process of blastulation, and blastomeres (bl) are set on the periphery. **B)** Blastula with blastoderm (bd) and blastocele (bc). **C)** Gastrulating blastula: blastopore (bp), blastocele (bc), archenteron (ac), endoderm (ed) and ectoderm (ec). **D)** 24 hpf gastrula. **E)** Embryo at 1.5 dpf with primordia of the oral tentacle (ot). **F)** 2 dpf embryo with developing oral tentacles and primitive gut tube (pgt).

[31], and *H. floridana* [16]. Although one was in the development stage, the analyzed samples indicate an asynchronous gonadal maturation that allows them to reproduce for several months during the reproductive season, as observed in Florida [32]. It is expected to find all reproductive stages during the reproductive months of similar species, such as *H. mexicana* in Panama [33] and Belize [34].

The variation in oocyte density between short and long tubules may indicate heterogeneity in the development of oogenesis among regions of the gonad and between short and large tubules. In *H. floridana*, females classified in stage III "Mature" exhibit tubules containing densely packed oocytes and a few early oocytes with visible germinal vesicles [16,30]. In this study, developing and mature oocytes were observed in the gonad classified at stage II "Development", indicating a clear oocyte development in the latter [30]. The apparent similarity in oocyte development in both short and long tubules and among sections of the tubules was less evident in spawned tubules since immature oocytes were also observed. Studies with other tropical holothuroids indicate that gametogenesis reinitiates in the spawned tubules, but the remanent oocytes are reabsorbed after the cessation of reproduction [28]. The histological results confirmed that the sea cucumbers used for spawning induction were mature and appropriate for reproduction.

The reproductive cycle of holothuroids is strongly influenced by environmental factors that affect their maturation and spawning, including seasonal variations, temperature, light intensity, and photoperiod, as well as the lunar cycle, salinity, tidal flow, and the availability and type of food, among other factors [28,35]. This relationship with the environment allows for various spawning induction methods when it does not occur spontaneously under captive conditions. These methods

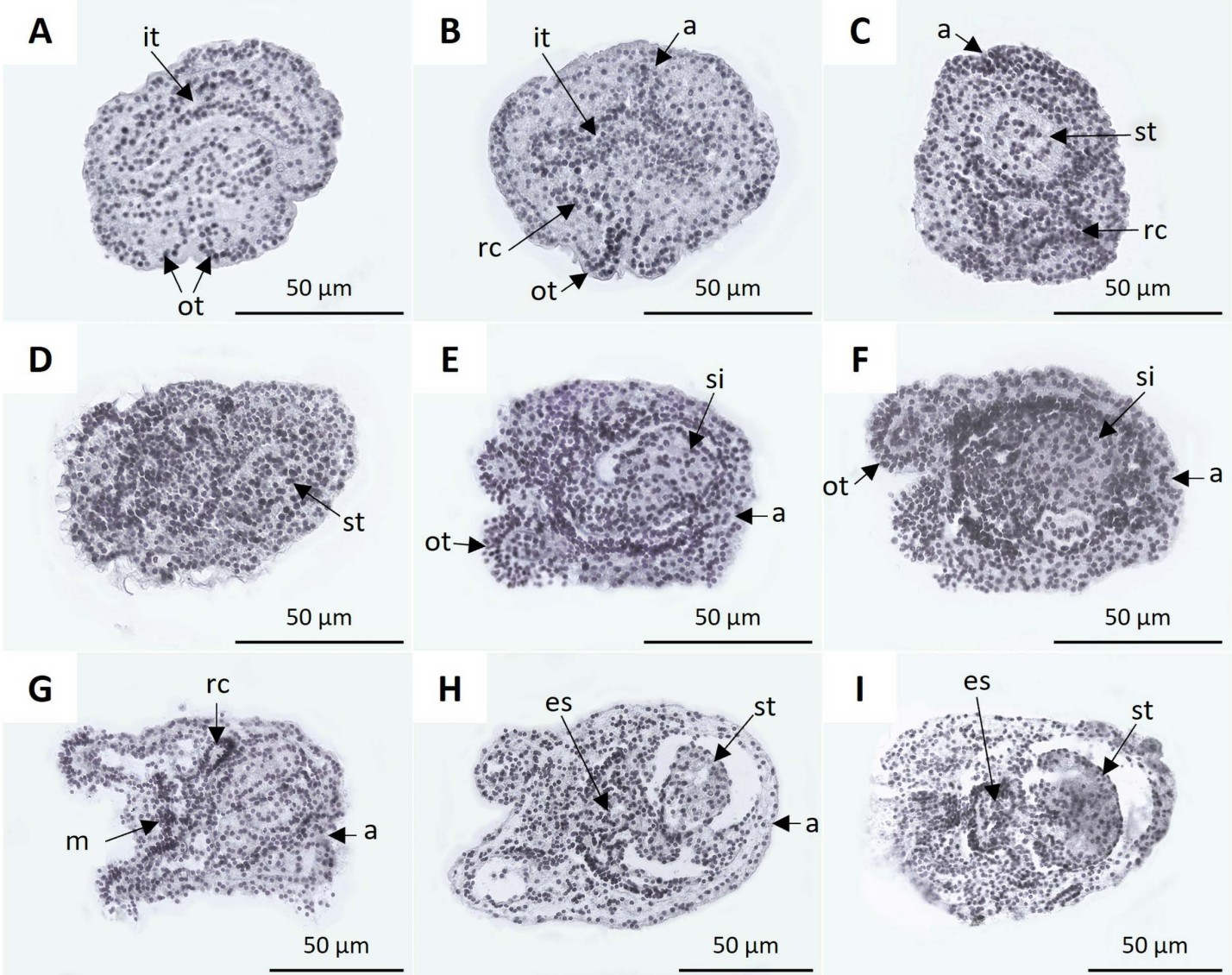

**Fig 8. Histological sections of vitellaria, pentactula, and juvenile of *H. floridana*. A)** 2.5 dpf larva with intestinal tubule (it) and oral tentacles (ot). **B)** Larva of 3 dpf with oral tentacles, intestinal tubule connected to the anus (a), and ring canal of the water vascular system (rc). **C)** larva of 3.5 dpf with stomach (st), anus, and ring canal. **D)** 4 dpf pentactula larva; the connection between the mouth and the stomach is observed. **E-F)** Pentactula at 4.5 and 5 dpf with sigmoidal intestine (si). **G)** 5.5 dpf pentactula larva with the mouth (m), anus, and ring canal. **H)** Pentactula of 6 dpf, showing the digestive system consisting of the esophagus(es), stomach, and anus. **I)** Juvenile of 7 dpf with esophagus and stomach.

include applying diverse stressors and using hormones or chemical substances that help gonad maturation and artificial spawning [19].

Successful reproduction induction techniques in sea cucumber species include desiccation in *H. leucospilota* [36] or microalgae stimulation in *H. fuscogilva* [37] and *H. sanctori* [38]. In this study, the serial thermal shock technique that stimulated the reproductive behavior and spawning of *H. floridana* was based on the method suggested for *H. scabra* [39], where 3–5°C increases the temperature more than that in the broodstock tanks. If it is higher than 30°C, the temperature is first decreased by 5°C; then the thermal shock is applied.

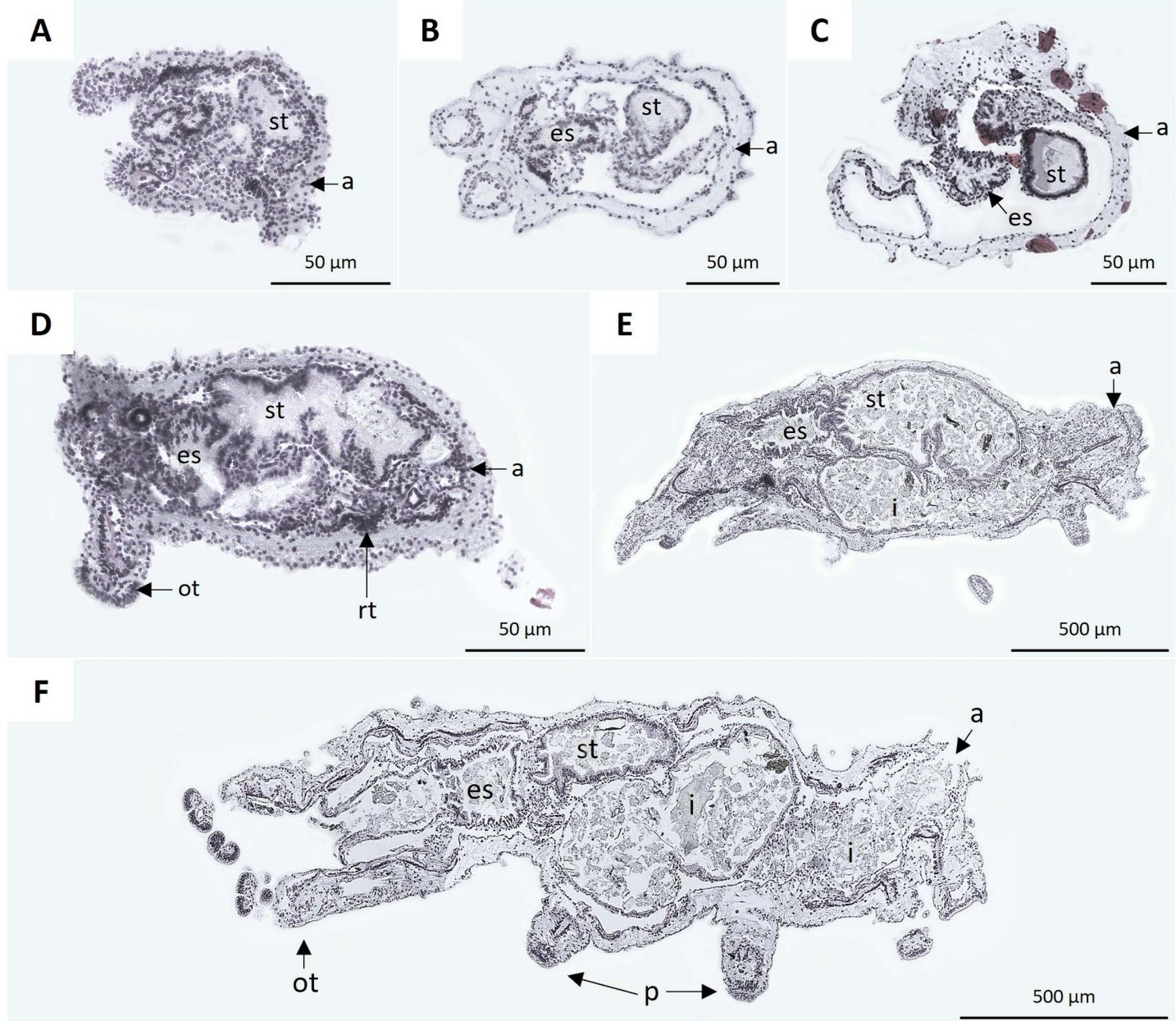

**Fig 9. Histological sections of *H. floridana* juveniles at 9 dpf (A): stomach (st) and anus (a). B)** Juvenile of 10 dpf: esophagus (es), stomach, and anus. **C)** Juvenile of 15 dpf: esophagus, stomach, and anus. **D)** Juvenile of 20 dpf with oral tentacles (ot), esophagus, stomach, anus, and respiratory tree (rt). **E)** Juvenile of 30 dpf: esophagus, stomach, intestine (i), and anus. **F)** Juvenile of 45 dpf: esophagus, stomach, intestine, anus, pelted oral tentacles, and podia (p).

In this study, however, lowering the temperature to 6–8°C was not enough to induce the spawning of *H. floridana*, since in Belize, spawning with this species was obtained when transferring the breeders from 34°C to 26°C [17]. The wide temperature range of 10–12°C used in this study can be attributed to the fact that this species lives in shallow waters subject to wide variations in environmental temperature, so it could need more extreme changes to stimulate spawning, as

observed in other tropical species. For example, the spawning of *Stichopus monotuberculatus* was stimulated by applying thermal shocks with a range of minimum temperatures of 8–10°C and maximum temperatures of 27°C [40]. In comparison, successful spawning of *H. fuscogilva* has been induced with a lowest temperature of 7°C and a maximum of 28°C [41].

This serial thermal shock, developed and applied in our laboratory during the last five years for *H. floridana*, is repeated as many as three times the same night until the desired number of animals spawned. Around 10% of the broodstock would spawn on any given induction night. However, we have obtained oocytes and sperm every time this protocol is performed, as long as it is applied to mature broodstock during the reproductive season, mainly within three days before and after the new or full moon [14].

Different studies pointed out that holothuroids with direct development, such as lecithotrophic larvae, produce large oocytes with diameters greater than 300 µm [19,42] as they contain the reserve substances necessary for the nutrition of the vitellaria larvae during their development. In the case of *H. floridana,* this development occurs within the fertilization envelope, with the larvae hatching as a pentactula rather than as a ciliated lecithotrophic planktonic larva such as seen in *Holothuria mexicana* [43]. The diameter of the oocytes of *H. floridana* in this study (394 µm on average) was larger than that reported previously [42], while in Belize were reported oocytes for the same species with a diameter of 265 µm preserved in 70% ethanol [17]. This variation in the size of different localities could be related to the age or size of the female and even to local environmental characteristics, which could influence embryonic development.

Given the non-planktonic lecithotrophic characteristic of the *H. floridana* vitellaria larva, fertilized oocytes settle on the substrate, where they adhere via the gelatinous capsule that encases them. This structure is not reported in other species of sea cucumber, and even previous studies with this species do not mention it in their descriptions of the larval development [18]. It was argued that this capsule allows the adhesion of oocytes on the body of the parents where embryonic development occurs [17]. However, the present study also demonstrates that the gelatinous capsule allows the oocyte to be covered with debris from the environment while remaining attached to the substrate, a strategy to go unnoticed by potential predators during embryonic development. The debris could also serve as a food source for the pentactula and early juvenile stages [17].

Early studies on the embryonic development of *H. floridana* reported that hatching occurs six days post-fertilization [20]. However, in this study, the breakage of the fertilization envelope was observed between 3–4 dpf, a time shorter than previously registered [17,18,20]. This difference may be related to the rearing temperature and appropriate environment for larval development under controlled conditions, which favor rapid development in aquaculture systems.

The first study related to the embryonic development of this species was made in 1909 [18], indicating that at the time of hatching, the pentactula larvae had four developed oral tentacles and a fifth short tentacle still in development, while in this study, it was observed that the pentactula larvae hatched with the five oral tentacles developed and with a smaller main podium, which was recently reported also [17]. In the present study, the appearance of ossicles was observed in the pentactula at 5 dpf, a time longer than the three days previously reported [18], when it is impossible to observe it because the fertilization envelope still covers the larvae. In this study, in the juvenile stage, the appearance time of the sixth oral tentacle at 45 dpf and the sighting of the oral tentacle buds developed and sectioned at 60 dpf were similar to those previously reported [18]. On the other hand, the respiratory tree identified in this research by histology at 20 dpf was previously reported at 40 dpf when fully developed, but with the possibility that they appeared in earlier stages [18], as observed in the present study.

This study confirms that *H. floridana* has an abbreviated lecithotrophic larval development, where the embryo undergoes a complex development within the fertilization envelope [17,20,44]. Unlike planktotrophic species, holothuroids with vitellaria lecithotrophic larvae bypass the auricularia and doliolaria phases and transition directly from the gastrula to a doliolaria-type larval form [19,44].

In this regard, no evidence was obtained by histology or SEM indicating the presence of cilia in the larva of *H. floridana* inside the fertilization envelope. The cilia of planktotrophic larvae are used for locomotion and feeding in the water column [44]. These characteristics are absent in the vitellaria larva of *H. floridana,* whose negatively buoyant oocytes sink and attach to the substrate via a gelatinous capsule. In contrast, *H. mexicana* larvae, which share the same habitat as *H. floridana*, exhibit free-swimming, ciliated lecithotrophic development, and reach the pentactula stage at 6 dpf [40,45].

Establishing the timeframe and characteristics when a pentactula becomes juvenile can be confusing. Other authors stated that the opening of the digestive tract through the mouth and anus, the appearance of the ossicles, the elongation of the primary podia, and the increase in the number of oral tentacles and podia are the attributes indicating the complete transition from pentactula to the early juvenile [43,44]. Based on these considerations, this study confirmed that the early juvenile stage of *H. floridana* occurred between 5 dpf, when the union of the stomach with the mouth and anus was completed, and the ossicles began to develop, and 7 dpf when the first secondary podium appeared.

In this study, it was observed that the development of the digestive system of *H. floridana* is similar, although in a shorter time, to that reported for *Eupentacta fraudatrix*, a lecithotrophic species of the order Dendrochirotida with shortened metamorphosis and without a radical transformation of the internal organs and tissues [46]. Both species coincided with the hatching time at 3–3.5 dpf [47]. In comparison, the results of this study indicate that *H. floridana* has a short early life cycle period in which the intestinal tubule has no connection to the blastoderm (~2 dpf) and that the connection with the mouth and anus is formed between 3 and 4 dpf, with evidence of a sigmoidal intestine between 4 and 5 dpf, which would indicate that the pentactula can feed at this point. The complete digestive system with the mouth, esophagus, stomach, intestine, and anus was observed in *H. floridana* at 6 dpf, quicker than the 9 dpf reported earlier [18].

The rapid lecithotrophic embryonic development of *H. floridana* is an advantage for its aquaculture as it does not require food and sophisticated infrastructure during this stage, reducing the costs involved in larviculture. *H. floridana* hatches as a pentactula larva around 4 dpf and becomes juvenile between 5 and 7 dpf, a very short time when compared to sea cucumber species with planktotrophic larvae as in the case of *A. japonicus*, whose larvae reach the pentactula stage at 16–17 dpf and juvenile at 30 dpf [48], or with *I. badionotus*, which is the most commercially important species in the Yucatan Peninsula and Greater Caribbean [7], whose free-swimming larvae reaches the pentactula stage after 19 dpf and the juvenile stage at 23–27 dpf [49].

The planktotrophic sea cucumber larvae aquaculture requires tanks that provide sufficient volume to develop swimming larvae. After settlement, it is necessary to change the settling substrates with the pentactula and juveniles to tanks with lower water-column but a larger area, as with *A. japonicus* [50]. On the other hand, *H. floridana*, having a benthic lecithotrophic development, requires only one type of tank with a larger area and low water column containing settling substrates, where the entire cycle takes place from embryo to juvenile. In addition, they do not require phytoplankton cultures to feed auricularia larvae but need the production of phytobenthos after the appearance of the pentactula larvae, which can be easily substituted with commercial microalgae concentrates. This characteristic represents cost savings in farm infrastructure for producing live food while reducing the risk of high mortality in the husbandry of planktonic larvae with long developmental periods, which requires specialized technical hatchery personnel.

## Conclusion

Based on the above, this research confirms and updates the knowledge that *H. floridana* exhibits an abbreviated lecithotrophic embryonic development. This species produces a non-ciliated vitellaria larva enclosed within a fertilization envelope, which develops attached to the substrate via a gelatinous capsule encasing the settling oocyte. Consequently, it does not undergo the swimming larval stages of auricularia and doliolaria, hatching as a pentactula larva at 4 dpf, and progresses to the early juvenile stage between 5–7 dpf. These characteristics confer significant technological advantages for its aquaculture compared to its planktotrophic counterparts. Its abbreviated development reduces the need for rearing and feeding swimming larval stages, which require sophisticated facilities and specialized technical expertise. Furthermore,

it eliminates the need for dedicated infrastructure, enabling the direct transfer of early juveniles to intermediate grow-out systems in the wild, facilitating their use in sea ranching or restocking programs.

In this sense, its direct embryonic development, the absence of planktotrophic larvae, and the possibility of inducing spawning through thermal shocks make *H. floridana* a species with high aquaculture potential. Its farming can be done in coastal areas within its geographical range in the tropical western Atlantic and the Greater Caribbean region, either in commercial aquaculture projects at different technological levels or for restoring overexploited wild populations.

## Supporting information

**S1 Dataset.**
(XLSX)

## Acknowledgments

The authors wan to recognize the Mexican Institute for Sustainable Fisheries and Aquaculture Research for its support through the staff from the Regional Center for Aquaculture and Fisheries Research at Yucalpetén, Yucatán, for collecting the broodstock, as well as the consulting company Aquatics Group for their logistical support.

## Author contributions

**Conceptualization:** Miguel A. Olvera-Novoa, Arlenie Rogers.

**Data curation:** Libni A. Maas-Hernández, Arlenie Rogers, Karina Macal-López.

**Formal analysis:** Libni A. Maas-Hernández, Arlenie Rogers, Karina Macal-López.

**Funding acquisition:** Miguel A. Olvera-Novoa.

**Methodology:** Libni A. Maas-Hernández, Miguel A. Olvera-Novoa, Arlenie Rogers, Luis Felaco, Teresa Colas-Marrufo.

**Resources:** Miguel A. Olvera-Novoa, Luis Felaco, Teresa Colas-Marrufo.

**Writing – original draft:** Miguel A. Olvera-Novoa.

**Writing – review & editing:** Libni A. Maas-Hernández, Miguel A. Olvera-Novoa, Arlenie Rogers, Luis Felaco, Karina Macal-López.

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
