## [Decision Letter · Decision Letter 0]

PONE-D-24-55150Description of the embryonic development of Holothuria floridana (Pourtalès, 1851) to produce juveniles for aquaculture and restokingPLOS ONE

Dear Dr. Olvera-Novoa,

Thank you for submitting your manuscript to PLOS ONE. After careful consideration, we feel that it has merit but does not fully meet PLOS ONE’s publication criteria as it currently stands. Therefore, we invite you to submit a revised version of the manuscript that addresses the points raised during the review process.

As both reviewers mention in their comments, this work presents numerous points that need to be improved before it can be considered for publication. In particular, Reviewer 2 raises several important critiques. One of these is the reproducibility of the experiments, including spawning percentages, among other aspects. Therefore, I recommend that the authors reproduce the experiments while addressing these critiques and include statistics and data on reproducibility so that the manuscript can be accepted. It is essential that these data appear in the revised version; otherwise, the article cannot be published.

We look forward to receiving your revised manuscript.

Kind regards,

Hector Escriva, PhD

Academic Editor

PLOS ONE

Journal Requirements:

3. In the online submission form, you indicated that The data supporting this study's findings are available from the corresponding author upon reasonable request

Reviewers' comments:

Reviewer's Responses to Questions

**Comments to the Author**

1. Is the manuscript technically sound, and do the data support the conclusions?

Reviewer #1: Yes

Reviewer #2: No

2. Has the statistical analysis been performed appropriately and rigorously? 

Reviewer #1: Yes

Reviewer #2: No

3. Have the authors made all data underlying the findings in their manuscript fully available?

Reviewer #1: Yes

Reviewer #2: Yes

4. Is the manuscript presented in an intelligible fashion and written in standard English?

Reviewer #1: Yes

Reviewer #2: No

5. Review Comments to the Author

Reviewer #1: In the submitted article entitled "Description of the embryonic development of Holothuria floridana (Pourtalès, 1851) to produce juveniles for aquaculture and restoking", the authors provide several pieces of knowledge regarding the development of the sea cucumber H. floridana from unfertilized eggs to 60 days after fertilization (DAF). Of particular interest to the community, this study depicts in great detail the embryonic development of this sea cucumber species, relying on brightfield images as well as histological investigations and scanning electron microscopy. H. floridana certainly presents a great interest in aquaculture. As such, the proposed study is important to provide a better understanding of the first steps of the development of this species and assess the quality of ongoing cultures.

Major points:

- The abstract currently describes the methods used in the study and known results from previous work rather than highlighting the study's key findings and conclusions. The authors should rewrite the abstract to help the reader grasp at first sight the context and interest of the study.

- The introduction lacks mention of knowledge from already published work on H. floridana, including the fact that histological studies on male and female tubules, as well as some images of fertilized oocytes, blastulae, and pentactulae, have already been published and provide some conclusions. This is an important piece of information that should be added to the introduction in addition to information on what the current study is adding as new knowledge.

- A table, histogram, or any kind of graphical representation would help visualize and understand the oocyte density distribution described in the result section "Histology of female gonads".

Other points:

- On page 11, the caption for Figure 3 is likely incorrect. Although B looks like it illustrates indeed the "first cell division until the formation of the two blastomeres", C likely shows an embryo after the second cell division, D after the third, and so on.

- On page 12, lines 243-244, the authors mention, for the first time, the presence of the intestine, while in Figure 5 (and its related caption), they indicate the presence of the intestine starting at 5 DAF. Please correct the statement or figure caption appropriately.

- On page 13, lines 268-271, the references to the images of Figure 7 should be corrected. As two examples, Fig 7 should read Fig 7A and Fig 7D should read Fig 7C.

- On page 15, in the caption for Figure 5, the authors should define what is the annular canal and pinpoint it in Figure 5C.

- On page 16, the authors mention on several occasions the presence of the intestine from juveniles 9 DAF to 20 DAF. It would be nice if they could provide an image showing the intestine at some of these stages in Figure 10.

- In the caption for Figure 5, the stomach is abbreviated "e" while in the captions for Figures 9 and 10, it is abbreviated "st". For consistency, please abbreviate similarly stomach through all figures (and the same applies to any other abbreviated terms).

Minor points:

- In the caption for Figure 5, the text "Juvenile of 20 DAF: sigmoid intestine and anus." should read "Juvenile of 20 DAF: sigmoid intestine and anus (a)."

- In the caption for Figure 8, "arquenteron" should read "archenteron" and "24-h" should read "24 HAF".

- In the caption for Figure 9, the text "ring canal of the water vascular system." should read "ring canal of the water vascular system (rc).

Reviewer #2: The manuscript “Description of the embryonic development of Holothuria floridana (Pourtalès, 1851) to produce juveniles for aquaculture and restoking" describes the reproductive biology and embryonic development of the sea cucumber H. floridana through histology and microscopy techniques, with the aim of designing protocols for reproduction and rearing of this species for aquaculture and restocking purposes.

The authors show H. floridiana development starting from the ovaries content and fertilized oocytes to the advanced juvenile stages, while the description of the efficacy of the spawning method is poorly presented. The authors do not mention how many times they have repeated the spawning induction and what was the overall success rate of spawning. This is an important point if the aim of the work is to develop protocols for aquaculture. In addition, the main developmental stages for this species were already described in previous works and the detailed description including all the histological observations are not indispensable to develop protocols for reproduction and rearing for aquaculture. On the other side, this detailed description could be useful for developmental biologists. The fast development and the absence of a planktonic larval stage before the formation of the juvenile, make this system potentially useful to study organogenesis and the evolution of pentameric structures in echinoderms. However, the authors made the description of development on embryos generated from oocytes deriving from one female, this making description and timing of the different stages presented in the figures and in Table 1 not robust.

For these reasons I consider the work incomplete and not worthy of publication in Plos One.

Some specific comments are provided below:

-The overall terminology used to describe the different phases of development is in many cases not appropriate and some sections are not clearly presented. Examples are the use of the word “organisms” when referring to the sea cucumber adults, embryos and larvae. Also, some headings in the methods sections are not appropriate, for examples “Embryonic development” and “Organogenesis” are more appropriate for a Results section and should be “Embryo cultures, fixation and imaging” and “Hystological analysis” respectively.

-Figures are not clear, adding more arrows could help the reader to identify the objects described in the text, for example: in Fig. 1 the authors refer to mature oocytes and degraded oocytes without pointing at them in the figures. This is true for most figures. Also, time of development in hpf and stages of development should be included in all the figures panels to help the reader.

Several English mistakes and wrong wording are present all over the manuscript, examples:

-lines 48-49: “Overfishing sea cucumbers has led to declining or disappearing wild populations, considering aquaculture as an alternative to meet its commercial demand” should be “Overfishing sea cucumbers has led to declining or disappearing wild populations, making aquaculture an alternative to meet its commercial demand”

-line 65: “They have an abbreviated lecithotrophic larval development” should be “They have a fast lecithotrophic larval development”

-line 111: “For this, samples of organisms were placed in a vial containing 2.5 mL of seawater and 2-3 drops of clove oil in seawater” should be “For each time point embryos, larvae and juveniles were placed in a vial containing 2.5 mL of seawater and 2-3 drops of clove oil in seawater”

-153: “weighed” should be “weighted”

-line 154 “Conformed” should be “formed”

-line 163: “vitellogenic oocytes were observed…” should be replaced with “oocytes”

-line 168-169 is not clear and should be refrased

-the abbreviations used for the statistics should be explained

-lines 206-210: the authors state that the adhesive gelatinous capsule gets thicker after fertilization, however this is not shown in the figure 2 and it is not presented in terms of results of the measurements (they only measure it after fertilization). They also refer to panel C of figure 2 to introduce the oocyte protuberance (that, by the way, they don’t define) while this is visible in panel B. In general, I don’t see figure 2 informative, it could be fused with figure 3.

-line 212: “sowing” should ne “showing”

-line 220: “A: formation of the polar body” is wrong, in the figure they show the extrusion of the polar body after fertilization; polar bodies form during oogenesis.

-HAF (hours after fertilization) and DAF (days after fertilization) should be replaced with the broadly utilized hpf and dpf (hors post fertilization and days post fertilization)

-line 241: “a main podia” should be “a main podium”

-line 284: the title “Organogenesis” should be “Embryonic development and organogenesis”

-line 297 “arquenteron” should be “archenteron”

-the Discussion section needs major English corrections

6. PLOS authors have the option to publish the peer review history of their article (what does this mean? ). If published, this will include your full peer review and any attached files.

**Do you want your identity to be public for this peer review?** For information about this choice, including consent withdrawal, please see our Privacy Policy .

Reviewer #1: No

Reviewer #2: No

---

## [Author Response · Author response to Decision Letter 1]

30 Jan 2025

RESPONSES TO THE EDITOR

1. Q: I recommend that the authors reproduce the experiments while addressing these critiques and include statistics and data on reproducibility so that the manuscript can be accepted.

2. A: Given that several sea cucumber hatchery manuals document the same methodology for other species (see the answer to Reviewer 2), the reproducibility of the method used here is achievable. For this reason, we did not consider repeating the experiment as a necessary condition for deeming this manuscript suitable for publishing.

3. Q: Please ensure that your manuscript meets PLOS ONE's style requirements, including those for file naming.

A: It was revised that the manuscript meets PLOS NONE’s style requirements

4. Q: In your Methods section, please provide additional information regarding the permits you obtained for the work. Please ensure you have included the full name of the authority that approved the field site access and, if no permits were required, a brief statement explaining why.

A: Thank you for this recommendation. We have made the necessary edits to include the government department that assisted us. Regarding permits, this kind of study does not require special licenses in Mexico, considering that we did not conduct experiments that compromised the life, health, or well-being of the broodstock we worked with. The broodstock was collected by personnel from the Mexican Institute for Research in Sustainable Fisheries and Aquaculture, the country's authority regulating this activity. The national authority provided the sea cucumbers for scientific purposes in aquaculture activities, with the consideration of returning it to the collection site at the end of the spawning season.

5. Q: In the online submission form, you indicated that The data supporting this study's findings are available from the corresponding author upon reasonable request

A: Thank you for this recommendation. We are sharing the data as supplementary information.

RESPONSES TO REVIEWERS

Reviewer 1

1. Q: The abstract currently describes the methods used in the study and known results from previous work rather than highlighting the study's key findings and conclusions. The authors should rewrite the abstract to help the reader grasp at first sight the context and interest of the study.

A: Thank you for this recommendation; we have rewritten the Abstract accordingly

2. Q: The introduction lacks mention of knowledge from already published work on H. floridana, including the fact that histological studies on male and female tubules, as well as some images of fertilized oocytes, blastulae, and pentactulae, have already been published and provide some conclusions. This is an important piece of information that should be added to the introduction in addition to information on what the current study is adding as new knowledge.

A: The introduction includes brief information related to the H. floridana female's gonads histology and maturation stages, but describing the histology and maturation cycle of the species was not an objective of this study. Considering that recent studies describe these in detail, we do not include a histology literature review in the introduction of this paper.

3. Q: A table, histogram, or any kind of graphical representation would help visualize and understand the oocyte density distribution described in the result section "Histology of female gonads"

A: Thank you for this recommendation. We did not include this update in the manuscript for the following reasons: a) Documenting female gonad maturation was not the main objective of this study; b) We documented female gonad ripeness to ensure that the oocytes were mature and viable for the study, and histology confirmed that the collected sea cucumbers were ripe enough to reproduce, and, c) Ramos Miranda et al. (2017 [16]) made a comprehensive summary of female gonad maturation, and we did not want to repeat what has already been published for H. floridana. However, we have included a table to clarify these results, indicating in Fig 1 the different phases observed.

4. Q: On page 11, the caption for Figure 3 is likely incorrect. Although B looks like it illustrates indeed the "first cell division until the formation of the two blastomeres", C likely shows an embryo after the second cell division, D after the third, and so on.

A: Captions have been edited accordingly.

5. Q: On page 12, lines 243-244, the authors mention, for the first time, the presence of the intestine, while in Figure 5 (and its related caption), they indicate the presence of the intestine starting at 5 DAF. Please correct the statement or figure caption appropriately.

A: The information in the text was corrected to match what is indicated in the figure.

6. Q: On page 13, lines 268-271, the references to the images of Figure 7 should be corrected. As two examples, Fig 7 should read Fig 7A and Fig 7D should read Fig 7C.

A: The missing information related to the description of the different images was included in the text.

7. Q: On page 15, in the caption for Figure 5, the authors should define what is the annular canal and pinpoint it in Figure 5C.

A: "Annular canal" was changed in the text to the correct denomination (ring canal) and included its acronym (rc) in the caption

8. Q: On page 16, the authors mention on several occasions the presence of the intestine from juveniles 9 DAF to 20 DAF. It would be nice if they could provide an image showing the intestine at some of these stages in Figure 10.

A: Numbers in the figures have changed. In Figure 9, images showing the intestine on days 9, 10, 15, 20, 30, and 45 after hatching are included, and the same is possible to observe in Figure 4. These were the only days when histology was made.

9. Q: In the caption for Figure 5, the stomach is abbreviated "e" while in the captions for Figures 9 and 10, it is abbreviated "st". For consistency, please abbreviate similarly stomach through all figures (and the same applies to any other abbreviated terms).

A: The abbreviation in Figure 4 (former 5) was changed for consistency with the now Figures 8 and 9

10. Q: In the caption for Figure 5, the text "Juvenile of 20 DAF: sigmoid intestine and anus." should read "Juvenile of 20 DAF: sigmoid intestine and anus (a)."

A: Thank you for the suggestion; it was corrected.

11. Q: In the caption for Figure 8, "arquenteron" should read "archenteron" and "24-h" should read "24 HAF".

A: Thank you for the suggestion; it was corrected.

12. Q: In the caption for Figure 9, the text "ring canal of the water vascular system." should read "ring canal of the water vascular system (rc).

A: Thank you for the suggestion; it was corrected.

Reviewer 2

1. Q: The authors show H. floridiana development starting from the ovaries content and fertilized oocytes to the advanced juvenile stages, while the description of the efficacy of the spawning method is poorly presented. The authors do not mention how many times they have repeated the spawning induction and what was the overall success rate of spawning. This is an important point if the aim of the work is to develop protocols for aquaculture.

A: The induction of spawning of H. floridana using thermal shock is a method widely applied worldwide with other sea cucumber species (see Agudo, 2006: https://www.aciar.gov.au/sites/default/files/legacy/node/758/Sandfish%20hatchery%20techniques%20%28english%29.pdf); Al Rashidi et al., 2012: https://scholar.google.com/citations?view_op=view_citation&hl=en&user=umXXeucAAAAJ&citation_for_view=umXXeucAAAAJ:W7OEmFMy1HYC; Altamirano & Rodriguez, 2022 https://www.researchgate.net/publication/363271266_Hatchery_production_of_sea_cucumbers_sandfish_Holothuria_scabra); Ito, 2018: https://www.ctsa.org/files/publications/SeaCucumberHatcheryManual.pdf).

This method has been regularly used in our lab and a hatchery in Panama to reproduce this species, at least during the last five years. It is applied to the selected broodstock on days coinciding with the new and full moon. Around 10% of treated broodstock respond to the stimulus in each induction, but the same female can spawn several times during the reproductive season (see Felaco et al., 2024: https://doi.org/10.1016/B978-0-323-95377-1.00008-4). Even though the induction was not an objective of this work, we have edited the section and added new information to answer this comment.

2. Q: In addition, the main developmental stages for this species have already been described in previous works, and the detailed description, including all the histological observations, is not indispensable to developing protocols for reproduction and rearing for aquaculture. On the other side, this detailed description could be useful for developmental biologists. The fast development and the absence of a planktonic larval stage before the formation of the juvenile make this system potentially useful for studying organogenesis and the evolution of pentameric structures in echinoderms. However, the authors made the description of development on embryos generated from oocytes deriving from one female, making the description and timing of the different stages presented in the figures and Table 1 not robust.

A: We do not know a description of the embryonic development previously published with the details that we are offering. Documenting female gonad maturation was not an objective of this study. Still, it was reviewed to confirm that the collected sea cucumbers were ripe enough to reproduce and ensure that the oocytes were mature and suitable for the study.

Besides that, our histological description gives those interested in the culture of this species a better understanding of the development of the different organs of interest, especially the digestive tract and respiratory tree. This information is essential to developing farming systems and protocols for handling and feeding pentactula and juveniles in the hatchery, giving the possibility of moving early stages towards intermediate growing systems directly to hapas in the wild as early stages as possible.

On the other hand, the embryonic development description will be the same using oocytes from one or several females. We do not expect differences if we use oocytes from one or various females or spawn or repeat this study several times using different females, which is time-consuming and expensive. The description was made using slides from several oocytes, all showing the same.

3. Q: The overall terminology used to describe the different phases of development is in many cases not appropriate and some sections are not clearly presented. Examples are the use of the word "organisms" when referring to the sea cucumber adults, embryos and larvae.

A: Thank you for the observation, which was attended to name the different stages accordingly.

4. Q: Also, some headings in the methods sections are not appropriate, for examples "Embryonic development" and "Organogenesis" are more appropriate for a Results section and should be "Embryo cultures, fixation and imaging" and "Hystological analysis" respectively.

A: Thank you for the suggestion; it was considered in the text when applicable.

5. Q: Figures are not clear, adding more arrows could help the reader to identify the objects described in the text, for example: in Fig. 1 the authors refer to mature oocytes and degraded oocytes without pointing at them in the figures. This is true for most figures. Also, time of development in hpf and stages of development should be included in all the figures panels to help the reader.

A: Thanks for your observation. The suggestions have been accounted for.

Several English mistakes and wrong wording are present all over the manuscript, examples:

6. Q: lines 48-49: "Overfishing sea cucumbers has led to declining or disappearing wild populations, considering aquaculture as an alternative to meet its commercial demand" should be "Overfishing sea cucumbers has led to declining or disappearing wild populations, making aquaculture an alternative to meet its commercial demand"

A: Thanks for your observation. Your suggestion was considered in the writing

7. Q: line 65: "They have an abbreviated lecithotrophic larval development" should be "They have a fast lecithotrophic larval development"

A: Thank you for the observation; the wording was modified accordingly.

8. Q: line 111: "For this, samples of organisms were placed in a vial containing 2.5 mL of seawater and 2-3 drops of clove oil in seawater" should be "For each time point embryos, larvae and juveniles were placed in a vial containing 2.5 mL of seawater and 2-3 drops of clove oil in seawater."

A: The suggestion was attended to.

9. Q: line 153: "weighed" should be "weighted"

A: Weighed is the correct expression for obtaining the weight of something

10. Q: line 154 "Conformed" should be "formed"

A: The observation was attended to.

11. Q: line 163: "vitellogenic oocytes were observed…" should be replaced with "oocytes"

A: The suggestion was agreed to.

12. Q: line 168-169 is not clear and should be refrased

A: The observation was attended to.

13. Q: the abbreviations used for the statistics should be explained

A: The suggestion was attended to.

14. Q: lines 206-210: the authors state that the adhesive gelatinous capsule gets thicker after fertilization, however this is not shown in figure 2 and it is not presented in terms of results of the measurements (they only measure it after fertilization). They also refer to panel C of figure 2 to introduce the oocyte protuberance (that, by the way, they don't define) while this is visible in panel B. In general, I don't see figure 2 informative, it could be fused with figure 3.

A: There was an error in writing “thicker” instead of “thinner.” As shown in Figure 2, the gelatinous capsule decays during embryonic development, and this process is described in the text.

Regarding Figure 2, omissions were attended, and that figure was merged with Figure 3 as suggested.

15. Q: line 212: "sowing" should ne "showing"

A: The observation was attended to.

16. Q: line 220: "A: formation of the polar body" is wrong, in the figure they show the extrusion of the polar body after fertilization; polar bodies form during oogenesis.

A: The observation was attended to.

17. Q: HAF (hours after fertilization) and DAF (days after fertilization) should be replaced with the broadly utilized hpf and dpf (hors post fertilization and days post fertilization)

A: The suggestion was attended to.

18. Q: line 241: "a main podia" should be "a main podium"

A: The suggestion was attended to.

19. Q: line 284: the title "Organogenesis" should be "Embryonic development and organogenesis"

A: The suggestion was attended to.

20. Q: line 297 “arquenteron” should be “archenteron”

A: The suggestion was attended to.

21. Q: the Discussion section needs major English corrections

A: Thanks for your comments on the English wording, which has been considered, mainly the typos. However, I would like to remark that the original and updated manuscript was extensively reviewed by a native English speaker, Dr. Arlenie Rogers, a researcher at the University of Belize and co-author of the article.

---

## [Decision Letter · Decision Letter 1]

PONE-D-24-55150R1Description of the embryonic development of Holothuria floridana (Pourtalès, 1851) to produce juveniles for aquaculture and restokingPLOS ONE

Dear Dr. Olvera-Novoa,

Thank you for submitting your manuscript to PLOS ONE. After careful consideration, we feel that it has merit but does not fully meet PLOS ONE’s publication criteria as it currently stands. Therefore, we invite you to submit a revised version of the manuscript that addresses the points raised during the review process.

Please correct the different minor point adressed by reviewer 1 before the final acceptation of your manuscript.

We look forward to receiving your revised manuscript.

Kind regards,

Hector Escriva, PhD

Academic Editor

PLOS ONE

Journal Requirements:

Reviewers' comments:

Reviewer's Responses to Questions

**Comments to the Author**

1. If the authors have adequately addressed your comments raised in a previous round of review and you feel that this manuscript is now acceptable for publication, you may indicate that here to bypass the “Comments to the Author” section, enter your conflict of interest statement in the “Confidential to Editor” section, and submit your "Accept" recommendation.

Reviewer #1: (No Response)

Reviewer #2: All comments have been addressed

2. Is the manuscript technically sound, and do the data support the conclusions?

Reviewer #1: Yes

Reviewer #2: (No Response)

3. Has the statistical analysis been performed appropriately and rigorously? 

Reviewer #1: Yes

Reviewer #2: (No Response)

4. Have the authors made all data underlying the findings in their manuscript fully available?

Reviewer #1: Yes

Reviewer #2: (No Response)

5. Is the manuscript presented in an intelligible fashion and written in standard English?

Reviewer #1: No

Reviewer #2: (No Response)

6. Review Comments to the Author

Reviewer #1: The authors have satisfactorily addressed my previous comments. However, while re-reading the manuscript, some additional minor points emerged and should be addressed before publication.

- In the results section entitled "Histology of female gonads" the authors should clarify the difference between "vitellogenic oocytes" and "mature oocytes" for non-specialists. If there is no difference between the two terms, the authors should always use the same term to avoid confusion. Also, in this section, the authors should indicate the type of oocytes they counted to determine the density and frequency of oocytes in the ovaries (i.e. vitellogenic oocytes only, mature oocytes only, or all oocytes including vitellogenic, mature, primary, and degrading oocytes).

- The beginning of the results section entitled "Embryonic development" includes some inconsistencies. From the text and the images provided, it sounds as if all images in Figure 1 correspond to fertilized oocytes and subsequent embryos. If so, Figure 1A illustrates indeed the fertilization envelope, but which surrounds the fertilized 1-cell stage embryo and not simply some yolk. Likewise, the protuberance seen in Figure 1B is at the level of the fertilization envelope and not the oocyte. The authors should revise their figure caption and results section based on this comment. Also, the figure caption has several typos that should be corrected.

- Still, in the results section entitled "Embryonic development", the authors should clarify, for non-specialists, the morphological difference between a morula and a blastula. This will help produce practical information for developing hatchery infrastructures. Also, the formation of a blastocoel is not conventionally accepted as a sign of invagination. The formation of a blastocoel is what characterizes the blastula stage. Invagination means the movement of tissues within the blastocoel, which is the mark of gastrulation and takes place as described here between 12 hpf and 24 hpf. The authors should thus correct their descriptions of these stages in the text and the figure caption. In Figure 3B, it is possible also that the embryo exhibits already a blastopore. If this is correct, this structure should be highlighted in the figure and mentioned in the text and the figure caption. Last, in the caption of Figure 3, the authors outline the presence of "primary oral tentacles", "oral tentacle primordia" and "defined oral tentacles" when, in the text, they first mention the presence of "primordia of the oral tentacles" and then "primary oral tentacles" but never refer to "defined oral tentacles". The authors should provide consistent naming of the tentacles according to their level of development and homogenize the text and figure caption accordingly.

- In Figure 4, the authors indicate in panel F the presence of an anus. This notion should be indicated in the text as well. Likewise, in the text, the authors highlight the increasing number of oral tentacles and secondary podia as a characteristic of the different juvenile stages they observed. They should pinpoint these tentacles and podia in Figure 4 panels G to I.

- On pages 18 and 19, in the paragraph discussing the histological observations, the authors write that, in their study, they observed a gonad in "the development stage" hence stage II and a gonad "classified at stage V" hence post-spawning. Yet, in the results section, the authors indicate that one of the gonads they studied was "in Stage III: Mature". The authors should revise this inconsistency.

- Still in the discussion, in lines 473-475, the authors indicate: "Unlike planktotrophic species, holothuroids with vitellaria lecithotrophic larvae do not proceed through the auricularia and doliolaria phases". Yet, in the sentence that just follows, they wrote: "they go directly from the gastrula to doliolaria-type larvae [19, 43]". This sounds incoherent. Maybe, there is a typo issue.

In addition to the above specific comments, several English incorrectness, typos, or lack of homogenization in Figure captions or in mentions of time post-fertilization are present all over the manuscript. Here are just some examples:

- In line 33, "aqueous vascular system" should read "water vascular system".

- In line 204, "in short and long ovary tubules and the sections anterior(A)," should read " in short and long ovary tubules and in anterior (A),".

- In line 229, please amend "with spawning" to "with egg spawning".

- In line 255, a verb is likely missing in the sentence "as the primordia of the oral tentacles".

- In line 366, the sentence "the histology of fertilized oocytes allowed the observation of" should read "the histology of samples post-fertilization allowed the observation of".

- In line 325, the caption title for Figure 7 should read "Histological sections of H. floridana embryos".

- In line 356, "At 20 days post-fertilization (dpf)" should read "At 20 dpf"

- In lines 481-482, the mention of "do not feed" is redundant with "lecithotrophic".

Reviewer #2: (No Response)

7. PLOS authors have the option to publish the peer review history of their article (what does this mean? ). If published, this will include your full peer review and any attached files.

**Do you want your identity to be public for this peer review?** For information about this choice, including consent withdrawal, please see our Privacy Policy .

Reviewer #1: No

Reviewer #2: No

---

## [Author Response · Author response to Decision Letter 2]

11 Apr 2025

RESPONSES TO THE EDITOR

1. We reviewed the reference list to ensure it was complete and accurate.

2. We added a new citation, including an article describing echinoderm oogenesis, to help address reviewers' concerns and included it in the reference list.

3. The numbering in the manuscript and the reference list was updated to reflect the inclusion of the new citation.

4. We attended to the reviewers' observations and recommendations, including our responses below.

RESPONSES TO REVIEWERS

Q1: In the results section entitled "Histology of female gonads" the authors should clarify the difference between "vitellogenic oocytes" and "mature oocytes" for non-specialists. If there is no difference between the two terms, the authors should always use the same term to avoid confusion. Also, in this section, the authors should indicate the type of oocytes they counted to determine the density and frequency of oocytes in the ovaries (i.e. vitellogenic oocytes only, mature oocytes only, or all oocytes including vitellogenic, mature, primary, and degrading oocytes).

R1: This research publication is intended for a specialized scientific audience familiar with the subject matter. As such, we do not consider it necessary to include definitions of technical terms within the text. However, for readers seeking clarification, we have provided a reference (Smily, 1990) where detailed definitions can be found. This is the reference:

Smiley, S. (1990). A review of echinoderm oogenesis. Journal of electron microscopy technique, 16(2), 93-114.

Q2: The beginning of the results section entitled "Embryonic development" includes some inconsistencies. From the text and the images provided, it sounds as if all images in Figure 1 correspond to fertilized oocytes and subsequent embryos. If so, Figure 1A illustrates indeed the fertilization envelope, but which surrounds the fertilized 1-cell stage embryo and not simply some yolk. Likewise, the protuberance seen in Figure 1B is at the level of the fertilization envelope and not the oocyte. The authors should revise their figure caption and results section based on this comment. Also, the figure caption has several typos that should be corrected.

R2: Thank you for this comment. We have revised and reworded the Figure 1 caption and moved the figure caption below the entire paragraph.

Q3: Still, in the results section entitled "Embryonic development", the authors should clarify, for non-specialists, the morphological difference between a morula and a blastula. This will help produce practical information for developing hatchery infrastructures. Also, the formation of a blastocoel is not conventionally accepted as a sign of invagination. The formation of a blastocoel is what characterizes the blastula stage. Invagination means the movement of tissues within the blastocoel, which is the mark of gastrulation and takes place as described here between 12 hpf and 24 hpf. The authors should thus correct their descriptions of these stages in the text and the figure caption.

In Figure 3B, it is possible also that the embryo exhibits already a blastopore. If this is correct, this structure should be highlighted in the figure and mentioned in the text and the figure caption.

Last, in the caption of Figure 3, the authors outline the presence of "primary oral tentacles", "oral tentacle primordia" and "defined oral tentacles" when, in the text, they first mention the presence of "primordia of the oral tentacles" and then "primary oral tentacles" but never refer to "defined oral tentacles". The authors should provide consistent naming of the tentacles according to their level of development and homogenize the text and figure caption accordingly. With regards to the comment on the definition of morula and blastula: to maintain the focus and conciseness of the manuscript, we have opted not to include basic definitions. Instead, an in-text citation has been provided, allowing readers to consult the referenced source for terminology clarification if needed.

R3: With regards to the comment on the formation of the blastocoel and gastrulation: we have edited the text to include this corrected information.

Regarding Figure 3B, we reviewed our files from all other embryonic development observations made in Mexico, Belize, and Panama and concluded that the blastopore does not appear at this stage. Regarding the captions in Figure 3, we have corrected the text and the captions accordingly.

Q4: In Figure 4, the authors indicate in panel F the presence of an anus. This notion should be indicated in the text as well. Likewise, in the text, the authors highlight the increasing number of oral tentacles and secondary podia as a characteristic of the different juvenile stages they observed. They should pinpoint these tentacles and podia in Figure 4 panels G to I.

R4: Thank you for this comment. We have edited the caption and text accordingly. Figure 4, panels G to I, were edited accordingly.

Q5: On pages 18 and 19, in the paragraph discussing the histological observations, the authors write that, in their study, they observed a gonad in "the development stage" hence stage II and a gonad "classified at stage V" hence post-spawning. Yet, in the results section, the authors indicate that one of the gonads they studied was "in Stage III: Mature". The authors should revise this inconsistency.

R5: Thank you for this comment. We have edited the text and the figure accordingly on page 10.

Q6: Still in the discussion, in lines 473-475, the authors indicate: "Unlike planktotrophic species, holothuroids with vitellaria lecithotrophic larvae do not proceed through the auricularia and doliolaria phases". Yet, in the sentence that just follows, they wrote: "they go directly from the gastrula to doliolaria-type larvae [19, 43]". This sounds incoherent. Maybe, there is a typo issue.

R6: Thank you for this comment. We have edited the text to make it coherent.

Q7:- In line 33, "aqueous vascular system" should read "water vascular system".

- In line 204, "in short and long ovary tubules and the sections anterior(A)," should read " in short and long ovary tubules and in anterior (A),".

- In line 229, please amend "with spawning" to "with egg spawning".

- In line 255, a verb is likely missing in the sentence "as the primordia of the oral tentacles".

- In line 366, the sentence "the histology of fertilized oocytes allowed the observation of" should read "the histology of samples post-fertilization allowed the observation of".

- In line 325, the caption title for Figure 7 should read "Histological sections of H. floridana embryos".

- In line 356, "At 20 days post-fertilization (dpf)" should read "At 20 dpf"

- In lines 481-482, the mention of "do not feed" is redundant with "lecithotrophic".

R7: Thank you for these comments. We have:

- Replaced the word “aqueous” with “water”

- Edited the text to read “in short and long ovary tubules and in anterior (A),".

- We did not amend the text to include “with egg spawning” as this paragraph speaks about spermiation in the same sentence and should, therefore, not include “eggs”.

- We have edited the text to fix the verb in the sentence “as the primordia of the oral tentacles”

- We edited the text to include "the histology of samples post-fertilization allowed the observation of".

- The caption of Figure 7 has been edited to include "Histological sections of H. floridana embryos".

- The text has been edited to include “At 20 dpf”

- The words “do not feed” have been removed from the sentence.

---

## [Editor Report · Decision Letter 2]

Description of the embryonic development of Holothuria floridana (Pourtalès, 1851) to produce juveniles for aquaculture and restoking

PONE-D-24-55150R2

Dear Dr. Olvera-Novoa,

We’re pleased to inform you that your manuscript has been judged scientifically suitable for publication and will be formally accepted for publication once it meets all outstanding technical requirements.

Kind regards,

Hector Escriva, PhD

Academic Editor

PLOS ONE
---

## [Editor Report · Acceptance letter]

PONE-D-24-55150R2

PLOS ONE

Dear Dr. Olvera-Novoa,

I'm pleased to inform you that your manuscript has been deemed suitable for publication in PLOS ONE. Congratulations! Your manuscript is now being handed over to our production team.

Kind regards,

on behalf of

Dr. Hector Escriva

Academic Editor

PLOS ONE